# CoME: Empowering Channel-of-Mobile-Experts with Informative Hybrid-Capabilities Reasoning

**Yuxuan Liu** [1 2 †]  **Weikai Xu** [3 2 †]  **Kun Huang** [2]  **Changyu Chen** [1 2 †]  **Jiankun Zhao** [2]  **Pengzhi Gao** [2]  **Wei Liu** [2]  **Jian Luan** [2]  **Shang Shuo** [4]  **Bo Du** [5]  **Ji-Rong Wen** [1 ‡]  **Rui Yan** [5 ‡]

## Abstract

Mobile Agents requires hybrid-capabilities reasoning, including screen summary, subtask planning, action decision and action function. However, existing agents struggle to achieve both decoupled enhancement and balanced integration of these capabilities. To address these challenges, we propose Channel-of-Mobile-Experts (**CoME**), a novel agent architecture consisting of four distinct experts, each aligned with a specific reasoning stage, CoME activates the corresponding expert to generate output tokens in each reasoning stage via *output-oriented activation*. To empower CoME with hybrid-capabilities reasoning, we introduce a progressive training strategy: **Expert-FT** enables decoupling and enhancement of different experts' capability; **Router-FT** aligns expert activation with the different reasoning stage; **CoT-FT** facilitates seamless collaboration and balanced optimization across multiple capabilities. To mitigate error propagation in hybrid-capabilities reasoning, we propose InfoGain-Driven DPO (**Info-DPO**), which uses information gain to evaluate the contribution of each intermediate step, thereby guiding CoME toward more informative reasoning. Comprehensive experiments show that CoME outperforms dense mobile agents and MoE methods on both AITZ and AMEX datasets.

## 1. Introduction

Mobile Agents can autonomously execute user instructions and have become a prominent research focus in both academia and industry. Their development is characterized by three major trends: (1) from *API invocation* (Wen et al., 2023a; Deng et al., 2024) to *action simulation* (Xu et al., 2025a; Li, 2021), enabling adaptation to more complex environment; (2) from *interactive exploration* (Lee et al., 2023; Wen et al., 2023b) to *supervised finetuning* (Zhang & Zhang, 2024; Cheng et al., 2024), allowing to solve more generalized instructions; (3) from *modular framework* (Wang et al., 2024a; Zhang et al., 2025; Liu et al., 2025a) to *holistic agent* (Chai et al., 2025; Lin et al., 2025; Gou et al., 2025), simplifying system design and training pipelines. Powered by advanced Multi-modal Large Language Models (MLLMs), mobile agents are undergoing a new paradigm shift from *end-to-end prediction* to *step-by-step reasoning*, which improves the accuracy and robustness of action decision (Wu et al., 2025; Xu et al., 2025b; Qin et al., 2025).

Achieving accurate action decisions remains challenging, even with Chain-of-Thought (CoT) prompting (Wei et al., 2022), as the reasoning process typically requires the agent to: perceive the current screen state, plan the next sub-task, generate high-level action decision and low-level action function (Zhang et al., 2024). This process requires multi-dimensional capabilities, which is referred to as **hybrid-capabilities reasoning** in Figure 1. However existing mobile agents either enhance individual capabilities on task-specific datasets (*e.g.,* screen understanding (Zhang et al., 2021; Wang et al., 2021; Li et al., 2021) or action grounding (Chai et al., 2025; Gou et al., 2025)), often lacking capabilities integration; or pre-train on large-scale dataset (Cheng et al., 2024; Wu et al., 2025; Qin et al., 2025), which can lead to unbalanced performance of different capabilities. Effective methods for decoupled enhancement and balanced integration of multiple capabilities are still lacking. Although Mixture-of-Experts (MoE) achieves partial capability decoupling by using input-oriented activation that routes different input tokens to different experts (Zhou et al., 2022; Jiang et al., 2024; Lieber et al., 2024; Team, 2024; Dai et al., 2024), the ideal hybrid-capabilities reasoning requires expert activation to align with the capabilities demanded to generate output tokens at each reasoning stage. Such output-oriented activation, however, is incompatible with the design of MoE.

† Work was done during the internship at Xiaomi. [1]Gaoling School of Artificial Intelligence, Renmin University of China [2]Xiaomi Inc. [3]Nanyang Technological University [4]Independent Researcher [5]Wuhan University. Correspondence to: Ji-Rong Wen <jrwen@ruc.edu.cn>, Rui Yan <rui.yan@whu.edu.cn>.

*Proceedings of the $43^{rd}$ International Conference on Machine Learning*, Seoul, South Korea. PMLR 306, 2026. Copyright 2026 by the author(s).

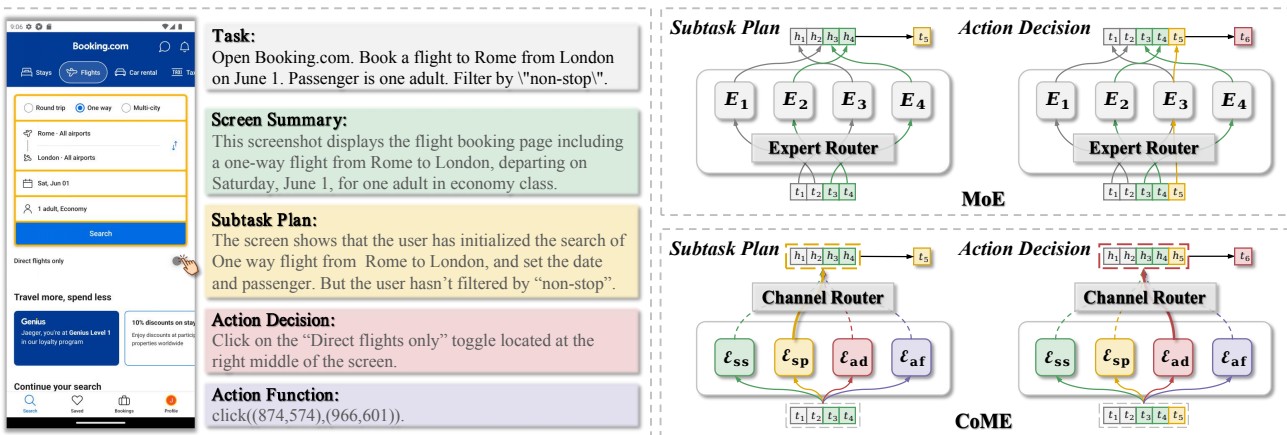

*Figure 1.* Left is an example of hybrid-capabilities reasoning. Right is the difference between input-oriented activation in MoE and output-oriented activation in CoME.

To address these challenges, we propose a novel agent architecture: **C**hannel-**o**f-**M**obile-**E**xperts (**CoME**), which incorporates four distinct experts to decouple hybrid-capabilities reasoning, each specialized in one of the capabilities: screen summary, subtask plan, action decision, and action function. In contrast to the input-oriented activation in MoE, CoME adopts output-oriented activation as shown in Figure 1. Specifically, CoME forwards the input tokens into each expert and selects the hidden states from the expert aligned with the current reasoning stage to generate each output token. To facilitate CoME with hybrid-capabilities reasoning, we design a progressive training strategy: (1) **Expert-FT** trains FFN layers on capability-specific data to initialize each expert, achieving effective decoupling and enhancement of different capabilities; (2) **Router-FT** trains channel router using expert label of each output token to align expert activation with reasoning stage; (3) **CoT-FT** trains CoME with the hybrid-capabilities reasoning data to enable seamless collaboration and balanced optimization among different experts. Through this progressive curriculum, CoME achieves hybrid-capabilities reasoning by activating specific expert aligned with reasoning stages.

As hybrid-capabilities reasoning spans multiple stages, even minor errors in intermediate steps can propagate and compromise the final outcome. To address this, we propose **InfoGain-Driven DPO (Info-DPO)**, which leverages information gain to quantify the contribution of each intermediate step to the final action prediction. Specifically, we use reward model to estimate the information entropy to the ground truth action before and after each reasoning stage, and use the reduction in entropy as InfoGain reward. Combined with action accuracy reward, we distinguish effective reasoning trajectories to construct high-quality DPO data. Through this mechanism, Info-DPO encourages the model to reinforce informative and reliable intermediate steps, while suppressing those contain distraction reason-

ing steps. Consequently, the model improves the reasoning accuracy at each stage and mitigates the error propagation throughout the reasoning trajectory. Experiments on the AITZ and AMEX datasets demonstrate that CoME outperforms dense mobile agents (+1.73%) and sparse MoE models (+5.72%) with equivalent activated parameters. Further analysis verifies that CoME achieves accurate expert activation aligned with reasoning stage, while Info-DPO improves the effectiveness of intermediate reasoning steps. Overall, our main contributions can be summarized as follows:

• We propose Channel-of-Mobile-Experts (**CoME**), a novel architecture incorporates experts specialized in: screen summary, subtask plan, action decision and action function. CoME employs output-oriented activation to activate appropriate expert at each stage of hybrid-capabilities reasoning.

• We develop a progressive training strategy to empower CoME for hybrid-capabilities reasoning: **Expert-FT** enables decoupling and enhancement of different capabilities to profile specific experts; **Router-FT** aligns expert activation with the reasoning stage; **CoT-FT** facilitates seamless collaboration and balanced optimization among experts.

• We introduce InfoGain-Driven DPO (**Info-DPO**), which uses information gain to guide preference construction by estimating the contribution of intermediate steps, to suppress invalid reasoning and mitigate error propagation. Comprehensive analysis demonstrates the effectiveness CoME.

## 2. Related works

### 2.1. Autonomous mobile agents

Mobile Agents can autonomously execute user instruction through API invocation or action simulation (Li et al., 2020a; Wen et al., 2023a; Deng et al., 2024; Liu et al., 2024), which have become a pivotal research spotlight. To equip the

agents with mobile knowledge, previous approaches introduce a prior exploration stage (Lee et al., 2023; Wen et al., 2023b; Li et al., 2024b), or design multiple specific tasks, such as screen understand (Zhang et al., 2021; Wang et al., 2021; Li et al., 2021), widgets recognition (Chen et al., 2022; Li et al., 2020b; Zheng et al., 2024), GUI transition (Gou et al., 2025; Wu et al., 2024a) and element/action grounding (Cheng et al., 2024; Chai et al., 2025; Bai et al., 2021; Baechler et al., 2024; Qian et al., 2024). Empowered by the advanced MLLMs, some research design some framework consists of multiple stages or agents (Wang et al., 2024a; Li et al., 2024b; Wang et al., 2025; Liu et al., 2025c). Recent works pre-train or finetune on large scale of mixed data to build general mobile agents (Zhang & Zhang, 2024; Huang et al., 2025; Xu et al., 2025b; Qin et al., 2025; Chen et al., 2025). GUI-R1 (Luo et al., 2025) and UI-S1 (Lu et al., 2025) enhance mobile agent generalization with rule-based RL. While existing mobile agents still struggle in capability disentanglement and balanced integration. Therefore, we propose CoME to facilitate hybrid-capabilities reasoning.

## 2.2. Mixture-of-Experts

Mixture-of-Experts (MoE) (Jacobs et al., 1991) integrates multiple experts and dynamically routes each input to the most relevant expert, to address diverse tasks (Shazeer et al., 2017; Lepikhin et al., 2021; Fedus et al., 2022). Recent works have integrated MoE into LLMs (Jiang et al., 2024; Lieber et al., 2024; Team, 2024; Dai et al., 2024) and MLLMs (Li et al., 2025; Deitke et al., 2025; Wu et al., 2024b) to boost capacity and efficiency by extending FFN layers into multiple experts and activating only top-K experts for each input token. AriaUI (Yang et al., 2025), the first MoE GUI agent, demonstrates the potential of MoE for mobile automation. However, MoE relies on input-oriented activation, which is not optimal for hybrid-capabilities reasoning. In contrast, CoME employs output-oriented activation to align expert activation with the reasoning stage.

# 3. Methodology

## 3.1. Task formulation

Given a user instruction $I$, at each step, the **Mobile Agent** $\mathcal{M}$ performs **hybrid-capabilities reasoning** conditioned on the current screen state $S$ and action history $H$ to generate the next action $a$. The reasoning trajectory $T$ comprises four distinct stages: (1) screen summary ($T_{ss}$) captures key screen information; (2) subtask planning ($T_{sp}$) identifies the next sub-goal; (3) action decision ($T_{ad}$) describes the chosen action; (4) action function ($T_{af}$) specifies the action type and parameters. The action $a$ is extracted from $T_{af}$.

$$a,\ T = \mathcal{M}(I, S, H),\ \text{where}\ T = [T_{ss}, T_{sp}, T_{ad}, T_{af}].\quad (1)$$

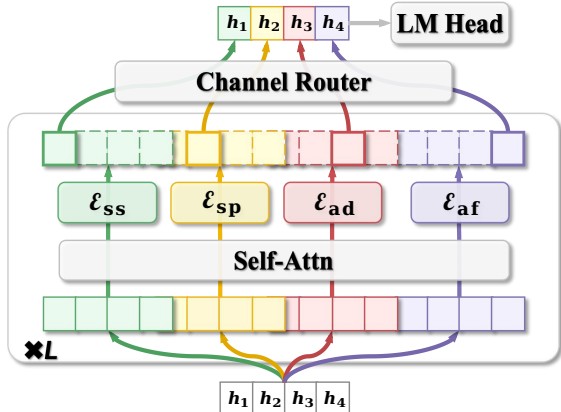

*Figure 2.* CoME architecture.

## 3.2. Channel-of-Mobile-Experts (CoME)

Mobile agents can automate user instructions through careful hybrid-capabilities reasoning, but still face two main challenges: dense models struggle to achieve decoupled enhancement and balanced integration of different capabilities, while MoE models fail to activate the expert aligned with the reasoning stage.

To address these challenges, we propose Channel-of-Mobile-Experts (CoME) in Figure 2, a novel agent architecture with *output-oriented activation*. CoME extends FFN module in each layer with four different experts $[\mathcal{E}_{ss}, \mathcal{E}_{sp}, \mathcal{E}_{ad}, \mathcal{E}_{af}]$, while shares the same Self-Attn module among experts. In contrast to the input-oriented activation in MoE[1], CoME adopts output-oriented activation to align the expert capability with the current reasoning stage. Specifically, given the hidden states $\mathbb{H} \in \mathbb{R}^{B \times N \times D}$ after the embedding layer, where $B$, $N$, $D$ stands for the batch size, sequence length and hidden size. CoME first repeats the hidden states for $E$ times $\mathbb{H} \rightarrow \widetilde{\mathbb{H}} \in \mathbb{R}^{B \times N \times E \times D}$, where $E$ represents the number of experts and $\widetilde{\mathbb{H}}$ represents the channel hidden states. At the $l$-th layer in CoME:

$$\begin{aligned}\widetilde{\mathbb{H}}^{(l)} &= \text{Self-Attn}\left(\widetilde{\mathbb{H}}^{(l-1)}\right) \\ \widetilde{\mathbb{H}}^{(l)}_{[e]} &= \text{FFN}_e\left(\widetilde{\mathbb{H}}^{(l)}_{[e]}\right), \quad e \in [1, \cdots, E]\end{aligned} \quad (2)$$

After obtaining the channel hidden state $\widetilde{\mathbb{H}}^{(L)}$ from the last layer, the most important step in CoME to achieve output-oriented activation is to select the hidden state from the corresponding expert channel according to the current reasoning stage. The channel router $\mathbf{W}_c \in \mathbb{R}^{ED \times E}$ will project the flattened hidden states into $E$-dimension channel logits $\mathbb{C} \in \mathbb{R}^{B \times N \times E}$, which will be used to fuse the hidden states from different expert channels to generate the final hidden states $\widehat{\mathbb{H}} \in \mathbb{R}^{B \times N \times D}$, achieving output-oriented activation.

---

[1]Limitation of MoE is analysed in Appendix A

Then $\widehat{\mathbb{H}}$ will be forwarded into LM-Head.

$$\mathbb{C} = \text{flatten}\big(\widetilde{\mathbb{H}}^{(L)}\big) \times \mathbf{W}_c,$$
$$\widehat{\mathbb{H}}_{[b,n]} = \sum_{e=1}^{E} \text{softmax}\big(\mathbb{C}\big)_{[e]} \cdot \widetilde{\mathbb{H}}_{[e]}^{(L)}, \quad (3)$$

### 3.3. Progressive training strategy

In order to empower CoME with hybrid-capabilities reasoning, we introduce a progressive training strategy: (1) **Expert Finetuning (Expert-FT)**, which explicitly decouples and enhances different capabilities; (2) **Router Finetuning (Router-FT)**, which allows expert activation aligned with the current reasoning stage; (3) **Chain-of-Thought Finetuning (CoT-FT)**, which facilitates seamless collaboration and balanced optimization among experts. Details in the following sections.

#### 3.3.1. STAGE-1: EXPERT FINETUNING (EXPERT-FT)

The hybrid-capabilities reasoning can be divided into four stages: screen summary, subtask plan, action decision and action function (Zhang et al., 2024), which is explicitly decoupled with different experts in CoME. To initialize and enhance these specialized experts, we train the FFN modules in Qwen2VL (Wang et al., 2024b) dense model on the dataset $\mathcal{D}_e$ of a specific ability respectively to initialize the different expert $e \in \big[\mathcal{E}_{\text{ss}}, \mathcal{E}_{\text{sp}}, \mathcal{E}_{\text{ad}}, \mathcal{E}_{\text{af}}\big]$

$$\mathcal{L}_{\text{Expert-FT}} = -\mathbb{E}_{\{x,y\}\sim\mathcal{D}_e}\big[\log \pi_{\mathcal{M}_e}\big(y \mid x\big)\big] \quad (4)$$

We extract the FFN layers in the specialized experts $\big[\mathcal{E}_{\text{ss}}, \mathcal{E}_{\text{sp}}, \mathcal{E}_{\text{ad}}, \mathcal{E}_{\text{af}}\big]$ to assemble CoME.

$$\text{FFN}_{\text{CoME}}^{(l)} = \Big[\text{FFN}_{\text{ss}}^{(l)}, \text{FFN}_{\text{sp}}^{(l)}, \text{FFN}_{\text{ad}}^{(l)}, \text{FFN}_{\text{af}}^{(l)}\Big]$$

#### 3.3.2. STAGE-2: ROUTER FINETUNING (ROUTER-FT)

After the Expert-FT, CoME has potentially mastered the capabilities required by hybrid-capabilities reasoning, thus it is necessary to enable output-oriented activation in CoME to align expert activation with the reasoning stage. We augment the hybrid-capabilities reasoning data $\mathcal{D}$ with the activated expert label of each output token according to the reasoning stage it is in. During training, we optimize the channel router using the Cross-Entropy loss $\mathcal{L}_{\text{R-CE}}$ between the predicted channel logits $\mathbb{C} \in \mathbb{R}^{B \times N \times E}$ and the expert labels $\mathcal{C} \in \mathbb{R}^{B \times N}$. To further supervise expert activation, we apply a Router Norm loss $\mathcal{L}_{\text{R-Norm}}$ as a regularization term to suppress irrelevant experts.

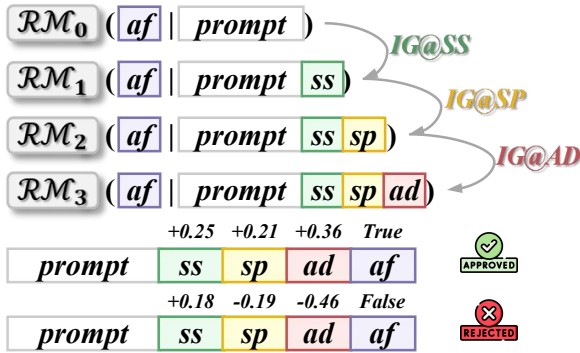

*Figure 3.* An example of InfoGain reward.

$$\mathcal{L}_{\text{R-CE}} = -\mathbb{E}_{\{\mathcal{C},\mathbb{C}\}\sim\mathcal{D}}\big[\mathcal{C} \cdot \log \text{softmax}\big(\mathbb{C}\big)\big]$$
$$\mathcal{L}_{\text{R-Norm}} = -\mathbb{E}_{\{\mathcal{C},\mathbb{C}\}\sim\mathcal{D}}\big[\|\text{softmax}\big(\mathbb{C}\big), \text{onehot}\big(\mathcal{C}\big)\|_2^2\big] \quad (5)$$
$$\mathcal{L}_{\text{Router-FT}} = \mathcal{L}_{\text{R-CE}} + \mathcal{L}_{\text{R-Norm}}$$

#### 3.3.3. STAGE-3: CHAIN-OF-THOUGHT FINETUNING (CoT-FT)

CoME achieves the decoupling and enhancement of multi-dimensional capabilities through Expert-FT, and aligns the expert activation with the reasoning stage through Router-FT. To further improve hybrid-capabilities reasoning, we design CoT Finetuning, which aims to simultaneously facilitates seamless collaboration and balanced optimization among different experts. During training, we use SFT loss $\mathcal{L}_{\text{SFT}}$ to optimize the hybrid-capabilities reasoning, and use a regularization term $\mathcal{L}_{\text{R-Norm}}$ to constrain expert activation during reasoning. We use $X$ to denote $\{I, S, H\}$.

$$\mathcal{L}_{\text{SFT}} = -\mathbb{E}_{\{X,T\}\sim\mathcal{D}}\big[\log \pi_{\mathcal{M}}\big(T \mid X\big)\big]$$
$$\mathcal{L}_{\text{CoT-FT}} = \mathcal{L}_{\text{SFT}} + \gamma \cdot \mathcal{L}_{\text{R-Norm}} \quad (6)$$

### 3.4. InfoGain-driven DPO

Chain-of-Thought stimulates the model's reasoning ability through step-by-step thinking to get more accurate results (Wei et al., 2022; Liu et al., 2025b). However, the intermediate reasoning steps may contain errors, which will impact the final accuracy because of the error propagation as the CoT length increases. In addition, reasoning trajectories that reach the correct answer via flawed steps should be suppressed, while those with logical steps but incorrect outcomes may still be beneficial. Therefore the key to improve the accuracy of reasoning is to ensure that each intermediate step has a positive contribution to the final answer. Since hybrid-capabilities reasoning is naturally a multi-stage reasoning, we can use information gain of each intermediate stage to measure its contribution (Ton et al., 2025).

$$IG(T_e) = \log \text{p}\big(T_{\text{af}} \mid X, T_{[0:e]}\big) - \log \text{p}\big(T_{\text{af}} \mid X, T_{[0:e-1]}\big)$$

*Table 1.* **Main results on AITZ dataset of different action.** Results with $^*$ are reported from (Zhang et al., 2024). Methods with $^\dagger$ are finetuned with hybrid-capabilities reasoning. The best overall result is marked **bold** and the second-best one is marked underline.

| Method | #Params | SCROLL | CLICK | | TYPE | | PRESS | STOP | Overall | |
|---|---|---|---|---|---|---|---|---|---|---|
| | | | type | match | type | match | | | type | match |
| Auto-GUI$^*$ (Zhang & Zhang, 2024) | 700M | 61.40 | 74.56 | 32.20 | 87.80 | 81.40 | 57.70 | 74.40 | **82.98** | 47.69 |
| Qwen2VL$^\dagger$ (Wang et al., 2024b) | 2B | 23.05 | 78.38 | 43.74 | 60.40 | 59.40 | 52.22 | 47.02 | 62.25 | 43.80 |
| ShowUI (Lin et al., 2025) | 2B | 23.79 | 53.29 | 40.34 | 89.60 | 85.20 | 59.00 | 87.55 | 64.22 | 49.55 |
| UITars (Qin et al., 2025) | 2B | 30.17 | 83.81 | 55.25 | 83.97 | 84.77 | 53.78 | 61.39 | 74.23 | 55.63 |
| GUI-R1 (Luo et al., 2025) | 3B | 25.53 | 79.00 | 50.47 | 21.80 | 20.80 | 32.89 | 28.77 | 57.32 | 40.41 |
| SeeClick (Cheng et al., 2024) | 7B | 11.14 | 69.92 | 52.96 | 53.80 | 53.00 | 67.88 | 55.36 | 62.93 | 49.11 |
| Qwen2VL$^\dagger$ (Wang et al., 2024b) | 7B | 43.28 | 83.97 | 55.30 | 51.80 | 51.40 | 56.92 | 64.87 | 71.59 | 54.46 |
| UGround (Gou et al., 2025) | 7B | 58.22 | 80.94 | 58.48 | 82.56 | 73.85 | 58.22 | 68.78 | 74.54 | 60.19 |
| OS-Atlas (Wu et al., 2025) | 7B | 76.12 | 75.82 | 54.83 | 89.80 | 88.60 | 68.67 | 81.75 | 77.83 | 65.11 |
| UITars (Qin et al., 2025) | 7B | 56.50 | 84.87 | 63.87 | 85.97 | 85.77 | 58.22 | 71.29 | 78.07 | 65.41 |
| GUI-R1 (Luo et al., 2025) | 7B | 29.02 | 82.22 | 53.88 | 29.60 | 27.00 | 45.17 | 42.26 | 62.87 | 45.91 |
| UI-S1 (Lu et al., 2025) | 7B | 45.60 | 78.75 | 56.73 | 91.20 | 87.20 | 51.17 | 39.28 | 69.85 | 56.22 |
| MolmoE$^\dagger$ (Deitke et al., 2025) | 1B | 28.19 | 79.84 | 33.17 | 65.20 | 62.00 | 37.07 | 54.96 | 65.96 | 38.23 |
| Qwen2VL-MoE$^\dagger$ | 3B | 48.75 | 80.03 | 59.43 | 77.80 | 74.00 | 57.70 | 70.83 | 72.94 | 60.69 |
| AriaUI (Yang et al., 2025) | 3.9B | 53.73 | 85.51 | 60.20 | 84.20 | 80.80 | 63.70 | 76.38 | 78.53 | 63.56 |
| DeepSeekVL2$^\dagger$ (Wu et al., 2024b) | 4.5B | 17.94 | 75.35 | 19.98 | 50.90 | 46.69 | 14.36 | 24.25 | 55.36 | 22.55 |
| **CoME** | 5B | 52.07 | 83.83 | 65.22 | 88.20 | 83.80 | 59.53 | 83.33 | 78.60 | **66.98** |

In order to estimate the information gain of the $e$-th reasoning stage, we train the reward model $\mathcal{RM}_e$, using the reasoning stages $T_{[0:e]}$ as additional inputs to directly predict the final action function $T_{\text{af}}$.

$$\mathcal{L}_{\mathcal{RM}_e} = -\mathbb{E}_{\{X,T\}\sim\mathcal{D}}\left[\log \pi_{\mathcal{RM}_e}\left(T_{\text{af}} \mid X, T_{[0:e]}\right)\right] \quad (7)$$

Therefore, the information gain of the $e$-th reasoning stage in Eq 7 can be approximated by Eq 8. Specifically, we denote the information gain in screen summary, subtask plan and action decision stages as *IG@SS*, *IG@SP* and *IG@AD* respectively as shown in Figure 3. Then we can calculate the reasoning-level InfoGain reward $\mathcal{R}_{\text{IG}} = IG@SS + IG@AP + IG@AD$ and the InfoPos reward $\mathcal{R}_{\text{IG}}^+ = \mathbb{1}_{IG@SS>0} \cdot \mathbb{1}_{IG@AP>0} \cdot \mathbb{1}_{IG@AD>0}$.

$$IG(T_e) \approx \log \frac{\pi_{\mathcal{RM}_e}\left(T_{\text{af}} \mid X, T_{[0:e]}\right)}{\pi_{\mathcal{RM}_{e-1}}\left(T_{\text{af}} \mid X, T_{[0:e-1]}\right)} \quad (8)$$

We use the action-level accuracy reward $\mathcal{R}_{\text{ACC}}$ in Eq 9. For click actions, $\mathcal{R}_{\text{ACC}}$ is computed based on the distance between the predicted and annotated coordinates within the threshold $\delta_d$; for input actions, it is the F1 score between the predicted and reference text within the threshold $\delta_f$; and for other actions, it is a binary exact-match indicator.

$$\mathcal{R}_{\text{ACC}} = \begin{cases} 1 - \max(\text{dis}(a, \hat{a}) / \delta_d, 1), & \text{click action} \\ \max(\text{f1}(a, \hat{a}), 0) \cdot \mathbb{1}_{\text{f1}>\delta_f}, & \text{input action} \\ \mathbb{1}_{a=\hat{a}}, & \text{other actions} \end{cases} \quad (9)$$

When generating DPO data, we sample $K$ reasoning trajectories for each action, and calculate the CoT-reward $\mathcal{R}_{\text{CoT}} = \mathcal{R}_{\text{IG}} \cdot \mathcal{R}_{\text{ACC}}$. We select the trajectory with the highest $\mathcal{R}_{\text{CoT}}$ as well as $\mathcal{R}_{\text{IG}}^+ = 1$ from the trajectories that hit the labeled action as the *chosen output*, which means that all of the reasoning stages are on the correct direction and lead to the most approximate answer to the labeled action. The *rejected output* is selected from the trajectories that don't hit the ground truth action and has the lowest $\mathcal{R}_{\text{IG}}$. More details about the DPO data selection can be found in Appendix B. After obtaining the DPO dataset $\mathcal{D}^*$, we can train CoME using the following DPO loss together with SFT loss $\mathcal{L}_{\text{SFT}}$ and Router Norm loss $\mathcal{L}_{\text{R-Norm}}$:

$$\mathcal{L}_{\text{IG-DPO}} = -\mathbb{E}_{\{X,T^+,T^-\}\sim\mathcal{D}^*}\left[\log \sigma\big(\beta(\Delta_{\mathcal{M}} - \Delta_{\mathcal{M}_{\text{ref}}})\big)\right]$$
$$+ \alpha \cdot \mathcal{L}_{\text{SFT}} + \gamma \cdot \mathcal{L}_{\text{R-Norm}} \quad (10)$$
$$\text{where} \quad \Delta_\theta = \log \frac{\pi_\theta(T^+|X)}{\pi_\theta(T^-|X)}$$

*Table 2.* **Main results on AMEX dataset of different apps**. Results with * are reported from (Chai et al., 2025). Methods with † are finetuned with hybrid-capabilities reasoning. The best overall result is marked **bold** and the second-best one is marked underline.

| Method | #Params | Gmail | Booking | Music | SHEIN | News | CM | ToDo | Signal | Yelp | Overall |
|---|---|---|---|---|---|---|---|---|---|---|---|
| Qwen2VL† (Wang et al., 2024b) | 2B | 37.4 | 35.2 | 40.1 | 33.7 | 50.0 | 47.5 | 41.8 | 56.7 | 40.5 | 38.53 |
| ShowUI (Lin et al., 2025) | 2B | 52.2 | 33.6 | 68.8 | 55.3 | 51.8 | 57.4 | 51.9 | 69.7 | 50.2 | 47.74 |
| UITars (Qin et al., 2025) | 2B | 59.4 | 47.9 | 55.4 | 53.0 | 65.1 | 62.3 | 64.7 | 61.7 | 58.1 | 54.43 |
| GUI-R1 (Luo et al., 2025) | 3B | 37.1 | 50.6 | 58.2 | 47.1 | 21.6 | 54.7 | 54.6 | 56.1 | 54.4 | 48.53 |
| SeeClick* (Cheng et al., 2024) | 7B | 28.2 | 29.4 | 18.1 | 20.0 | 30.0 | 53.1 | 30.7 | 37.1 | 27.4 | 30.44 |
| Qwen2VL† (Wang et al., 2024b) | 7B | 57.6 | 58.4 | 56.5 | 47.3 | 64.2 | 66.3 | 60.9 | 72.8 | 54.8 | 57.99 |
| UGround (Gou et al., 2025) | 7B | 70.9 | 68.8 | 72.7 | 63.7 | 77.7 | 67.7 | 73.7 | 80.1 | 67.6 | 69.12 |
| OS-Atlas (Wu et al., 2025) | 7B | 74.4 | 69.7 | 74.6 | 64.0 | 80.7 | 63.6 | 65.3 | 83.3 | 62.3 | 69.99 |
| SphAgent* (Chai et al., 2025) | 7B | 61.7 | 68.2 | 77.7 | 72.0 | 71.9 | 64.6 | 79.6 | 71.3 | 69.6 | 70.71 |
| UITars (Qin et al., 2025) | 7B | 67.7 | 70.0 | 71.8 | 63.8 | 71.5 | 67.7 | 77.0 | 86.4 | 72.8 | 70.33 |
| GUI-R1 (Luo et al., 2025) | 3B | 36.8 | 53.0 | 64.6 | 49.4 | 20.6 | 68.6 | 64.5 | 68.5 | 62.3 | 52.47 |
| UI-S1 (Lu et al., 2025) | 7B | 47.3 | 62.2 | 66.0 | 54.1 | 38.1 | 75.3 | 72.7 | 80.2 | 63.6 | 60.57 |
| MolmoE† (Deitke et al., 2025) | 1B | 38.7 | 28.6 | 28.1 | 27.2 | 45.3 | 22.8 | 37.6 | 38.2 | 34.0 | 31.56 |
| Qwen2VL-MoE† | 3B | 65.1 | 63.7 | 57.8 | 63.0 | 78.0 | 60.3 | 70.2 | 76.5 | 61.1 | 64.56 |
| AriaUI (Yang et al., 2025) | 3.9B | 63.1 | 62.3 | 68.5 | 58.9 | 83.0 | 54.7 | 62.5 | 83.3 | 66.9 | 64.10 |
| DeepSeekVL2† (Wu et al., 2024b) | 4.5B | 43.1 | 36.9 | 52.5 | 42.2 | 42.7 | 51.1 | 47.0 | 61.7 | 38.9 | 42.22 |
| **CoME** | 5B | 76.2 | 72.6 | 81.0 | 64.3 | 81.2 | 63.2 | 72.6 | 78.4 | 66.9 | **72.61** |

# 4. Experiment

## 4.1. Experiment setup

We train and evaluate CoME on two datasets, **AITZ** (Zhang et al., 2024) and **AMEX** (Chai et al., 2025). We compare against 13 baselines, covering both dense mobile agents and sparse MoE models, and report the accuracy of action type and match. More details on datasets, baselines, metrics, and implementation are provided in the Appendix D.

## 4.2. Main results

**AITZ.** As shown in Table 1, CoME achieves the highest overall action match accuracy. Compared to dense mobile agents,CoME yields an improvement of 11.35% over 2B-series model and 1.57% over 7B-series models, with only 5B activated parameters. Moreover, CoME surpasses MoE-based models by 3.42%. Baselines generally underperform on the CLICK action, while CoME achieves the highest accuracy of 65.22% (+1.45%), because CLICK is a representative hybrid-capabilities task that is much more challenging. Baselines also show imbalanced performance across action types (*e.g.,*, ShowUI: 87.55% STOP vs. 23.79% SCROLL). In contrast, CoME achieves a higher relative improvement (+11.56%) and lower bias (4.41) in Table 11, indicating more balanced action performance.

**AMEX.** As shown in Table 2, CoME achieves the best overall performance across nine apps, surpassing the dense model (+1.90%) and the sparse MoE (+8.05%). Compared with OS-Atlas, SphAgent and UITars pre-trained with large scale mobile data then finetuned on AMEX, CoME could

*Table 3.* Ablation analysis on AITZ and AMEX.

| Method | AITZ | AMEX | AVG |
|---|---|---|---|
| CoME | 66.98 | 72.61 | 69.78 |
| - w/o Info-DPO | 62.93 | 67.28 | 65.10 |
| - w/o Router-FT | 60.05 | 62.00 | 61.02 |
| - w/o Expert-FT | 57.07 | 64.47 | 60.74 |
| - Info-DPO w/o $\mathcal{L}_{\text{R-Norm}}$ | 65.96 | 70.90 | 68.42 |
| - CoT-FT w/o $\mathcal{L}_{\text{R-Norm}}$ | 62.57 | 64.24 | 63.40 |

surpasses them by 2.26% on average, only using AMEX data through hybrid-capabilities reasoning, which proves that CoME can better activate multi-capabilities to achieve more effective CoT reasoning.

## 4.3. Ablation analysis

We designed comprehensive ablation experiments to analyze the effects of different training stages and strategies. As shown in Table 3, Info-DPO contributes most to the action prediction (+4.68%), proving that using information gain to distinguish reasoning trajectories can mitigate error propagation and improve reasoning accuracy. Further, removing Router-FT leads to an accuracy decline (-4.08%), indicating that Router-FT enables expert activation aligned with the reasoning stage. Moreover, no prior Expert-FT can not fully release the expert's specialized capability (-4.36%). Including router norm $\mathcal{L}_{\text{R-Norm}}$ on expert activation is necessary in CoT-FT (+1.70%) and Info-DPO (+1.36%) as well.

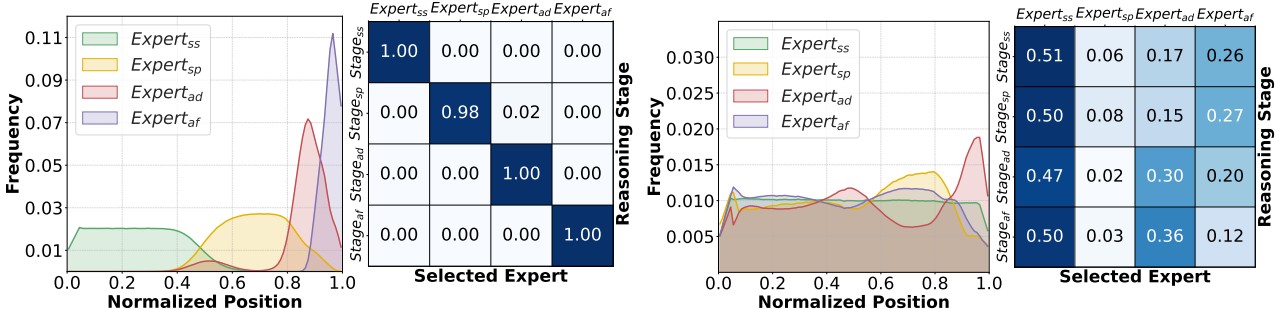

*Figure 4.* Expert distribution of CoME (left) and MoE (right).

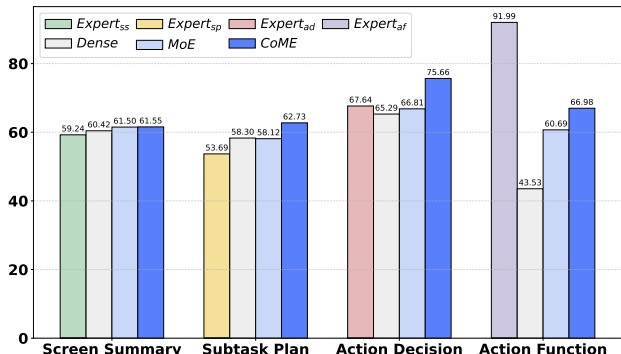

*Figure 5.* Performance on different reasoning stages.

### 4.4. Comprehensive analysis of CoME design

#### 4.4.1. ANALYSIS OF COME ARCHITECTURE.

***Q1: How CoME performs across different capabilities?***
CoME integrates four specialized experts: screen summary ($\mathcal{E}$ss), subtask plan ($\mathcal{E}$sp), action decision ($\mathcal{E}$ad), and action function ($\mathcal{E}$af). We compare CoME with Dense and MoE model, and use specific-capability expert as reference baseline in Figure 5. Dense and MoE method struggle in later reasoning stages, because the first two stages (screen summary and subtask plan) account for 80% of the tokens, which dominate CoT training and leave later stages inadequately trained. Conversely, by activating the expert aligned with reasoning stage, **CoME enables CoT-FT to optimize each expert in its specialized stage, resulting in balanced improvements across all capabilities.** The superior performance of action-function expert is because of leveraging ground-truth action decisions, whereas others use their own predicted decisions. By disentangling and fine-tuning each expert for its specific capability, CoME strengthens intermediate reasoning and effectively mitigates error propagation.

***Q2: Why CoME improves on hybrid-capabilities reasoning?*** We compare CoME against MoE by analyzing expert activation distributions and selection accuracy at each stage. As shown in Figure 4, expert activation in CoME exhibits clear stage preference (e.g., $\mathcal{E}_{ss}$ for initial screen

*Table 4.* Comparison on Android Control unseen splits.

| Method | app_un | task_un | cate_un | average |
|---|---|---|---|---|
| Qwen2VL (2B) | 51.25 | 51.94 | 50.60 | 51.26 |
| UITars (2B) | 58.90 | 60.21 | 58.59 | 59.23 |
| Qwen2VL (7B) | 63.19 | 64.65 | 63.68 | 63.84 |
| OSAtlas (7B) | 63.56 | 64.98 | 63.58 | 64.04 |
| CoME (5B) | **65.29** | **67.23** | **65.84** | **66.12** |

*Table 5.* Performance on AMEX auxiliary GUI tasks.

| Model | Caption | OCR | Grounding | Action |
|---|---|---|---|---|
| Qwen2VL (7B) | **54.53** | 78.87 | 78.89 | 57.99 |
| Ultars (7B) | 54.08 | 78.87 | 79.11 | 70.33 |
| MoE (A3B) | 52.34 | 78.75 | 77.32 | 64.56 |
| CoME (5B) | 53.89 | **79.67** | **79.19** | **72.61** |

summary and $\mathcal{E}_{af}$ for final action function) and achieves 99% selection accuracy, whereas MoE activation remain uniformly distributed and poorly aligned with reasoning stages. The experiment demonstrates that **CoME achieves output-oriented activation in hybrid-capabilities reasoning that successfully activate the expert with corresponding capability required by the reasoning stage.**

***Q3: How CoME performs on OOD settings?*** Table 4 evaluates CoME on three out-of-distribution (OOD) splits of Android Control (Li et al., 2024a): unseen apps (*app_un*), unseen tasks (*task_un*), and unseen categories (*cate_un*). CoME achieves the best performance over all splits, reaching 66.12 on average and surpassing OSAtlas (7B) by 2.08 points with a smaller 5B backbone. OOD scenarios require stronger transferable and compositional reasoning abilities. By aligning expert activation with different output reasoning stages, CoME better coordinates specialized capabilities across different reasoning stages. As a result, **CoME achieves more robust OOD performance**.

***Q4: How CoME performs on other GUI tasks?*** Beyond action prediction, we further examine whether CoME can activate appropriate experts for other GUI understanding tasks.

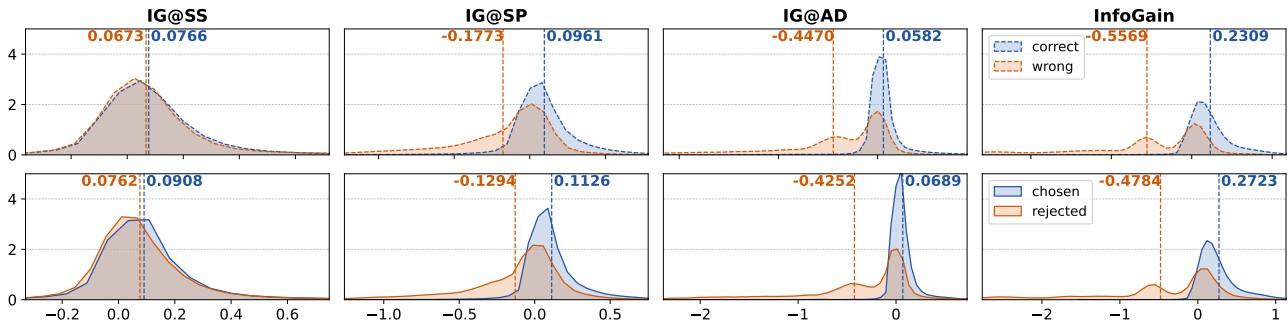

*Figure 6.* Information gain distribution. Above is the sampled data. Below is the DPO data. We illustrate the information gain at screen summary, subtask plan and action decision stage, as well as the total information gain of the entire reasoning trajectory.

*Table 6.* CoME variants with different numbers of experts.

| #Expert | scroll | click | type | press | stop | overall |
|---------|--------|-------|------|-------|------|---------|
| 3-expert | 41.22 | 61.52 | 79.80 | 66.57 | 71.31 | 62.28 |
| 4-expert | 52.31 | 62.42 | 71.80 | 63.37 | 69.24 | 62.93 |
| 8-expert | 54.39 | 61.96 | 79.20 | 56.92 | 73.02 | 63.59 |

*Table 7.* Comparison of DPO Strategies.

| Method | scroll | click | type | press | stop | overall |
|--------|--------|-------|------|-------|------|---------|
| **AITZ** | | | | | | |
| CoT-FT | 51.74 | 61.77 | 70.00 | 63.44 | 68.84 | 62.25 |
| Rule-DPO | 51.98 | 62.26 | **86.20** | **61.80** | 80.36 | 65.37 |
| Info-DPO | **52.07** | **65.22** | 83.80 | 59.53 | **83.33** | **66.98** |
| **AMEX** | | | | | | |
| CoT-FT | 70.83 | 66.39 | 79.16 | 70.66 | 51.14 | 67.28 |
| Rule-DPO | 82.34 | 66.31 | 78.38 | 48.00 | 58.35 | 70.20 |
| Info-DPO | **82.43** | **68.09** | **82.58** | **70.67** | **67.74** | **72.61** |

*Table 8.* Analysis of data pairing strategies.

| Strategy | InfoGain | InfoPos | Action Acc |
|----------|----------|---------|------------|
| cc+cw+lw | -0.50 / 0.26 | 0.03 / 0.61 | 77.60 / 65.96 |
| cc+cw | -0.48 / **0.27** | 0.03 / **0.70** | 78.60 / **66.98** |
| cw+lw | -0.71 / 0.26 | 0.04 / 0.46 | 76.88 / 64.07 |
| cw | -0.71 / 0.27 | 0.04 / 0.56 | 77.75 / 65.93 |

Specifically, we construct three auxiliary task categories on the AMEX test set, including Caption, OCR, and Grounding. As shown in Table 5, CoME achieves competitive or superior performance compared with the main baselines across these tasks. These results suggest that **CoME is not limited to improving action prediction through hybrid-capabilities reasoning, but also learns transferable GUI understanding capabilities that can generalize to diverse GUI-centric tasks.**

***Q5: Does CoME support different architecture?*** To validate the flexibility and scalability of CoME, we evaluate a 3-expert variant by removing the screen-summary expert and an 8-expert variant by doubling the number of experts at each reasoning stage. As shown in Table 6, **CoME supports different expert configurations and can be adjusted flexibly**, with action accuracy improving as the number of experts increases. More Details in Appendix F.

### 4.4.2. ANALYSIS OF INFO-DPO

***Q6: How Info-DPO performs compared with Rule-DPO?*** We compare Info-DPO with Rule-based reward DPO (Rule-DPO) in Table 7. Experimental results shows that Info-DPO surpasses Rule-DPO on both AITZ (+1.61%) and AMEX (+2.41%), and outperforms Rule-DPO across all action categories. For click actions, Rule-DPO yields marginal overall improvements (+0.51% and –0.08%), whereas Info-DPO achieves substantial gains (+3.45% and +1.70%). Because click action is highly sensitive to intermediate error: an incorrect element description can lead to large coordinate deviations. Unlike Rule-DPO with no supervision over intermediate reasoning, **Info-DPO employs information**

gain to identify and reinforce valid intermediate steps, thereby mitigating error propagation.

***Q7: How InfoGain-reward performs on reasoning trajectory evaluation?*** Figure 6 presents the Information gain distribution for the sampled data and DPO data. As reasoning progresses, the difference between correct and wrong reasoning increases (0.0766 / 0.0673 → 0.0582 / -0.4252). This confirms that the **InfoGain-reward can evaluate and distinguish the reasoning trajectory, where correct reasoning brings positive InfoGain while wrong reasoning results in negative InfoGain**. Furthermore, filtering DPO data with InfoGain-reward raises the information gain of the selected trajectories, demonstrating its effectiveness in removing invalid intermediate steps.

***Q8: How to select effective data for Info-DPO?*** For each action, we sample ten reasoning trajectories and construct

Table 9. Analysis of reward strategies.

| Method | InfoGain | InfoPos | Action |
|---|---|---|---|
| Reward-7B | 0.9841 | 0.6610 | 78.60 / 66.98 |
| Reward-72B | 0.8365 | 0.4533 | 78.52 / 66.85 |
| Continuous | 0.9841 | 0.6610 | 78.60 / 66.98 |
| Discrete | 0.7541 | 0.5018 | 77.94 / 65.51 |

Table 10. Efficiency analysis of CoME architecture.

| Method | GPU Mem (GB) | | Accuracy (%) | |
|---|---|---|---|---|
| | train | infer | type | match |
| Qwen2VL (7B) | 31.62 | 16.52 | 71.59 | 54.46 |
| UITars (7B) | 32.08 | 16.92 | 78.07 | 65.41 |
| MoE (A3B) | 30.70 | 22.32 | 72.94 | 60.69 |
| CoME (5B) | 18.52 | 11.69 | 78.60 | 66.98 |

DPO pairs using one of three strategies: (1) **cc**: pair two correct reasoning trajectories; (2) **cw**: pair a correct reasoning trajectory with a wrong one; and (3) **lw**: pair the label reasoning trajectory with an wrong reasoning trajectory. Details in Appendix B. Experimental results in Table 8 and Table 13 show that, higher InfoGain (informative contribution) and InfoPos (intermediate validity) leads to higher action accuracy. While, including label reasoning trajectories as chosen outputs (lw) provides no gain because of the distribution gap; introducing suboptimal correct reasoning as rejected outputs (cc) further mitigates invalid intermediate steps. Thus, **the most effective pairing strategy is to pair trajectories with high information gain as chosen outputs and those exhibiting incorrect outcomes or invalid intermediate steps as rejected outputs**.

### 4.4.3. ANALYSIS OF EFFICIENCY

***Q9: Which reward strategy is better for Info-DPO?*** We analyze the impact of different reward strategies on Info-DPO, including the scale of the reward model and the design of the reward calculation function. As shown in Table 9, using a **7B reward model can achieve comparable results of 72B reward model and even better**. Because the 72B reward model has stronger fitting ability and lower prediction loss, which makes the entropy difference between different reasoning stages smaller. Since InfoGain is calculated based on such differences, an overly strong reward model may produce weaker and less distinguishable InfoGain signals, thereby reducing its effectiveness for constructing preference pairs. Moreover, we find that **a continuous reward function in the range of 0–1 is more effective than a discrete 0/1 reward function**. This advantage is particularly important for actions such as CLICK and TYPE, where pre-

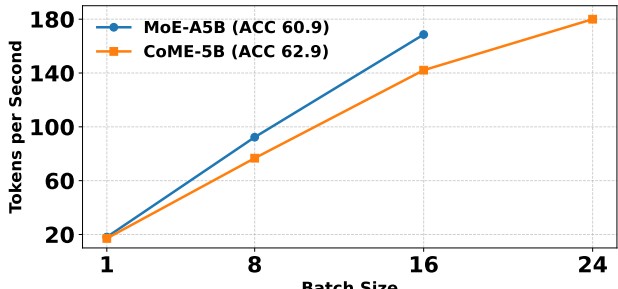

Figure 7. Comparison of batch inference between CoME and MoE.

diction quality is not purely binary. The detailed action-level results in Table 14 further support this observation.

***Q10: How efficient is CoME?*** We compare CoME with both dense and MoE model in Table 10, **CoME achieves better action accuracy while maintaining lower GPU memory usage**. GPU memory usage is dominated by model parameters, while the KV cache during inference contributes only a small fraction. More details in Appendix J. We also compare the inference efficiency of CoME against MoE-A5B, and report the throughput under different batch sizes. As shown in Figure 7, CoME achieves comparable inference speed to MoE, with only a minor slowdown of up to 1-1.25×. Since CoME has a lower memory footprint, it can scale to larger batch sizes under the same GPU memory budget, thereby compensating for the slight per-batch slowdown and achieving comparable overall throughput to MoE.

## 5. Conclusion

In this work, we propose Channel-of-Mobile-Experts (CoME), a novel mobile-agent architecture with output-oriented expert activation for hybrid-capabilities reasoning. By aligning expert activation with different reasoning stages, CoME enables specialized experts to handle screen summary, subtask planning, action decision, and action-function invocation. We further develop a progressive training curriculum, including Expert-FT, Router-FT, and CoT-FT, to achieve decoupled capability enhancement and balanced expert collaboration. To reduce error propagation in intermediate reasoning, we introduce InfoGain-Driven DPO, which derives a process-level reward from the information gain of each reasoning step. This reward identifies intermediate steps that contribute more to the final action decision, guiding DPO to reinforce reasoning that better supports task completion. Experiments on AITZ and AMEX show that CoME consistently outperforms dense and MoE baselines. Comprehensive analyses further verify the effectiveness of its architecture and training strategies. These findings suggest that aligning expert activation with reasoning stages offers a promising direction for building stronger, more flexible, and more generalizable mobile GUI agents.

## Acknowledgments

This work is supported by Xiaomi. This work is also supported by the Public Computing Cloud, Renmin University of China, and by the fund for building world-class universities (disciplines) of Renmin University of China.

## Impact statement

We introduce hybrid-capabilities reasoning in mobile scenarios, in which a model, given the current state, need to proceed through multiple reasoning stages to arrive at an action decision—each stage potentially requiring a different capability. Such multi-stage reasoning is ubiquitous in agent systems. When an agent decides on an action, it typically engages in environment perception, task planning, and action-function generation, all of which exemplify hybrid-capabilities reasoning. Consequently, the Channel-of-Mobile-Experts architecture proposed in this paper can be applied to a wide range of agent tasks, and our training strategy can be employed to train diverse agent applications.

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

## A. Limitation of Mixture-of-Experts on hybrid-capabilities reasoning

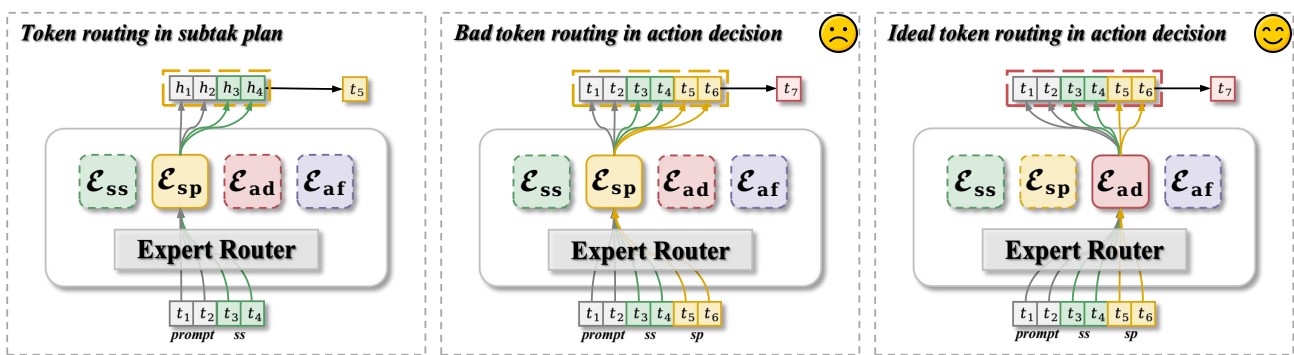

*Figure 8.* A case to show the limitation of MoE on hybrid-capabilities reasoning.

Mixture-of-Experts (MoE) (Jacobs et al., 1991) architectures have gradually gained traction in LLMs. By introducing multiple FFN modules as experts alongside a routing/gating mechanism, MoE sparsely activates only a subset of the experts for each token in the input sequence, thereby enabling an increase in total parameter while keeping computational costs relatively manageable (Zhou et al., 2022; Jiang et al., 2024; Dai et al., 2024). By assigning different subsets in the input to different experts, MoE implicitly links distinct capabilities to some specific experts, thereby achieving capability disentanglement in hybrid-capability scenarios. However, this represents an input-oriented activation, which is better for handling multi-task or multi-modal inputs.

Mobile Agents present an even more complex scenario. In order to achieve accurate action prediction, the agent need to perceive the current screen state, plan the next sub-task, then generate the high-level action decision and the low-level action function. This process involves multi-dimensional reasoning capabilities, which is referred to as hybrid-capabilities reasoning. Ideally, we want to activate the experts with the corresponding capability conditioned on the current reasoning stage to generate the output token, hence, this represents an output-oriented activation.

The reason why MoE can not achieve output-oriented activation can be attributed to the auto-regressive mechanism, which forces the same token in the input sequence routed to the same expert when generating tokens in different reasoning stages. As shown in Figure 8, although we can force the input tokens in prompt and screen summary to be routed to the specialized expert $\mathcal{E}_{\text{sp}}$ to generate the output token in the subtask plan stage, when generating output tokens in action decision stage, these input tokens will be forwarded to the $\mathcal{E}_{\text{sp}}$ again instead of the $\mathcal{E}_{\text{ad}}$. Thus, we can not use $\mathcal{E}_{\text{ad}}$ to process these preceding tokens, preventing the fully leveraging of capability of $\mathcal{E}_{\text{ad}}$ in action decision.

## B. DPO data selection

To construct DPO data pairs, we sample $K$ reasoning trajectories $\mathcal{T}$ for each action. These trajectories fall into three categories, and we adopt a tailored pairing strategy for each using the reward $\mathcal{R}_{\text{IG}}$, $\mathcal{R}_{\text{IG}}^{+}$, $\mathcal{R}_{\text{ACC}}$ and $\mathcal{R}_{\text{CoT}}$:

**The sampled trajectories are all correct.** We partition the sampled trajectories by whether $\mathcal{R}_{\text{IG+}} = 1$. From the $\mathcal{R}_{\text{IG+}} = 1$ subset, we select the trajectory with the highest $\mathcal{R}_{\text{CoT}}$ as the chosen output, which represents the most effective reasoning trajectory leading to the correct result. Conversely, from the $\mathcal{R}_{\text{IG+}} = 0$ subset, we pick the trajectory with the lowest $\mathcal{R}_{\text{IG}}$ as the rejected output, to suppress the trajectory that reaches a correct answer while through invalid intermediate reasoning steps.

$$\text{chosen: } \arg\max\big\{\mathcal{R}_{\text{CoT}}(T^{(k)}) \,|\, T^{(k)} \in \mathcal{T}, \mathcal{R}_{\text{IG}}^{+}(T^{(k)}) = 1\big\}$$
$$\text{rejected: } \arg\min\big\{\mathcal{R}_{\text{IG}}(T^{(k)}) \,|\, T^{(k)} \in \mathcal{T}, \mathcal{R}_{\text{IG}}^{+}(T^{(k)}) = 0\big\} \tag{11}$$

**The sampled trajectories are partially correct.** We partition the sampled trajectories by whether $\mathcal{R}_{\text{ACC}} > 0$. From the $\mathcal{R}_{\text{ACC}} > 0$ subset, we prioritize selecting the trajectory with the highest $\mathcal{R}_{\text{CoT}}$ and $\mathcal{R}_{\text{IG+}} = 1$ as the chosen output, which represents the most effective reasoning trajectory leading to the correct result. From the $\mathcal{R}\text{ACC} = 0$ subset, we select the trajectory with the lowest $\mathcal{R}\text{IG}$ as the rejected output, as it represents the worst reasoning and should be avoided.

$$\text{chosen:} \begin{cases} \arg\max\{\mathcal{R}_{\text{CoT}}(T^{(k)}) \mid T^{(k)} \in \mathcal{T}, \mathcal{R}_{\text{ACC}}^+(T^{(k)}) > 0, \mathcal{R}_{\text{IG}}^+(T^{(k)}) = 1\} \\ \qquad \text{if}\{T^{(k)} \mid \mathcal{R}_{\text{IG}}^+(T^{(k)}) = 1\} \neq \phi \\ \arg\max\{\mathcal{R}_{\text{CoT}}(T^{(k)}) \mid T^{(k)} \in \mathcal{T}, \mathcal{R}_{\text{ACC}}^+(T^{(k)}) > 0, \mathcal{R}_{\text{IG}}^+(T^{(k)}) = 0\} \\ \qquad \text{if}\{T^{(k)} \mid \mathcal{R}_{\text{IG}}^+(T^{(k)}) = 1\} = \phi \end{cases} \tag{12}$$
$$\text{rejected:} \ \arg\min\{\mathcal{R}_{\text{IG}}(T^{(k)}) \mid T^{(k)} \in \mathcal{T}, \mathcal{R}_{\text{IG}}^+(T^{(k)}) = 0\}$$

**The sampled trajectories are all wrong.** We use the ground truth reasoning trajectory $T^*$ as chosen output and choose the trajectory with the lowest $\mathcal{R}$IG as the rejected output.

$$\text{chosen:} \ T^*$$
$$\text{rejected:} \ \arg\min\{\mathcal{R}_{\text{IG}}(T^{(k)}) \mid T^{(k)} \in \mathcal{T}\} \tag{13}$$

## C. Progressive training strategy

In order to empower CoME with hybrid-capabilities reasoning, we propose a progressive training strategy consists of three stages: (1) **Expert Finetuning (Expert-FT)**, which explicitly decouples and enhances different capabilities; (2) **Router Finetuning (Router-FT)**, which allows the activation of expert aligned with the current reasoning stage; (3) **Chain-of-Thought Finetuning (CoT-FT)**, which facilitates seamless collaboration and balanced optimization among experts. The overall training framework is shown in Figure 9.

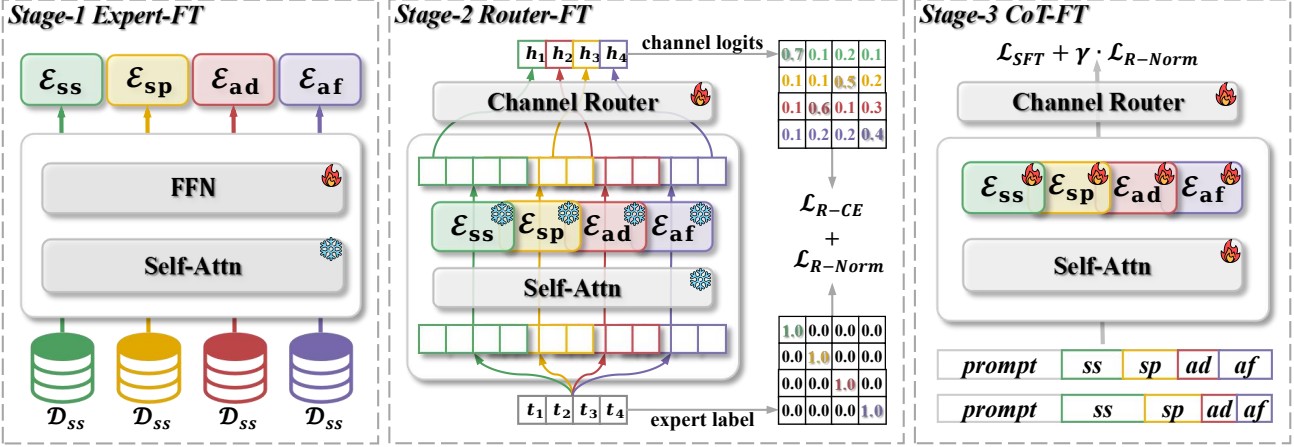

*Figure 9.* Overall training framework of CoME.

## D. Experiment setup details

### D.1. Dataset

We train and evaluate CoME on both AITZ (Zhang et al., 2024) and AMEX (Zhang et al., 2024).

**AITZ** is a cleaned subset of the large scale AITW (Rawles et al., 2023), comprising 2.5K unique instructions and 18K steps. These instructions, drawn from over 70 mobile apps, are divided into five categories: general, google apps, install, web shopping, and single. Each step is annotated with a chain-of-action-thought, covering screen summarization, action planning, action description, and action result.

**AMEX** is a large-scale mobile dataset comprising 104k screenshots and 3k instructions collected from 64 distinct apps. It features approximately 1.6M GUI element grounding annotations and 712k GUI element function descriptions. We further augment AMEX using the data construction methodology established by AITZ, the augmented data will be open-sourced.

### D.2. Baselines

We compare CoME with two types of baselines: (1) Dense mobile agents and (2) Sparse MoE models.

**Dense mobile agents** are the general mobile agents pre-trained on large mobile datasets with strong capability on mobile environments, including:

• **Auto-GUI** (Zhang & Zhang, 2024) proposes a chain-of-action technique, that incorporates the text described previous action history and future action plan to facilitate action decision on the current screen.

• **SeeClick** (Cheng et al., 2024) only relies on screenshots for mobile task automation. SeeClick is pre-trained on large scale of mixed data including widget caption, UI summarization and UI grounding, together with general multi-modal dataset, *e.g.,* VQA and visual reasoning.

• **ShowUI** (Lin et al., 2025) uses UI-guide visual token selection to formulate screenshot as an UI connected graph, thus reducing the computational cost. The introduced interleaved vision-language-action streaming can flexibly manage visual-action history in navigation to enhance training efficiency.

• **UITars** (Qin et al., 2025) is a native GUI agent that solely perceives the screenshot as input and perform end-to-end action decision, which incorporates the unified action modeling and system-2 reasoning. UITars is pre-trained on large-scale screenshots with precise caption and grounding annotation. Moreover, it is iterative trained with reflective online traces to continuously learn from its mistakes.

• **UGround** (Gou et al., 2025) is trained with a large scale of 10M GUI elements and the target bounding box over 1.3M screenshots to empower strong grounding ability on mobile environment.

• **OS-Atlas** (Wu et al., 2025) is a foundational GUI action model with grounding mode, action mode and agent mode. It can generate an action description as a simple plan and grounding the action on the screenshot, which excels at GUI grounding and OOD agentic tasks.

• **SphAgent** (Chai et al., 2025) is pre-trained on the AMEX dataset, including three-level annotations: interactive element grounding, screen and element functionality descriptions, and instruction with action chains. SphAgent is equipped with strong capabilities on screen understanding and element recognition.

• **GUI-R1** (Luo et al., 2025) is a strong reasoning-centric mobile GUI agents. It leverages explicit step-by-step reasoning (R1-style) to plan actions and then executes the predicted operation.

• **UI-S1** (Lu et al., 2025) is trained with Semi-online Reinforcement Learning, which simulates online rollouts using offline trajectories to better capture multi-step signals without costly environment interaction.

**Sparse MoE models** are the MoE base models that are pretrained on the general visual-language tasks and fine-tuned on the mobile environment by us, including:

• **MolmoE** (Deitke et al., 2025) is a multi-modal MoE MLLM with 1.5B active and 7.2B total parameters, which nearly matches the performance of GPT-4V on both academic benchmarks and human evaluation.

• **DeepSeekVL2** (Wu et al., 2024b) incorporates a dynamic tiling vision encoding strategy to process high-resolution images and leverages DeepSeekMoE (Dai et al., 2024) with Multi-head Latent Attention and shared experts, which has only 4.5B activated parameters.

• **AriaUI** (Yang et al., 2025) is the first native MoE-based mobile agent that leverage textual or text-image interleaved action from trajectory to enhance the dynamic contexts understanding.

### D.3. Metrics

Following Auto-GUI (Zhang & Zhang, 2024), we use **action type** score to evaluate the accuracy of the predicted action type as well as the **action match** score to measure the accuracy of the predicted action parameter. For CLICK action, the distance between predicted action and labeled action less than 140 on the normalized 0–1000 coordinate scale is considered correct, or both of the clicked coordinate fall in the same bounding box. For TYPE action, the F1 score between predicted text and labeled text greater than 0.5 is considered correct. For other actions, we use exact match between the predicted action and labeled action.

### D.4. Implementation details

CoME comprises four distinct experts, each initialized from the Feed-Forward Network (FFN) layers of Qwen2VL-2B. During the Expert-FT stage, we exclusively fine-tune the FFN layers while freezing all other parameters. Training is performed on task-specific subsets—screen summary, subtask plan, action decision, and action function—extracted from the original hybrid-capabilities reasoning data of AITZ and AMEX. This stage is conducted using a cosine learning rate scheduler with a peak learning rate of 1e-5. In the Router-FT stage, we annotate each output token according to its corresponding reasoning stage and expert. Only the channel router is trained in this stage, using a peak learning rate of 1e-4 with a cosine scheduler. During the CoT-FT stage, we fine-tune the language model component of CoME while freezing the Vision Transformer (ViT). We adopt a LoRA configuration with rank 64 and LoRA alpha 128. This stage uses a peak learning rate of 2e-4 and follows a cosine learning rate schedule. For Info-DPO, we sample $K = 10$ reasoning trajectories for each step. And regarding the thresholds in Eq. 9, we set the distance threshold $\delta_d$ to 50 and the function threshold $\delta_f$ to 0.5. During Info-DPO training, we maintain the same configuration as CoT-FT but reduce the peak learning rate to 1e-4. For the weights of auxiliary SFT loss and Router Norm loss in Eq. 6 and Eq. 10, we set $\alpha$ to 1 and $\beta$ to 0.1.

## E. Analysis of relative improvement results on AITZ

We conducted a more detailed analysis of the performance improvements across all methods. Specifically, we computed the average performance of all methods and treated it as a new baseline (Average). We then measured each method's improvement relative to this baseline. Additionally, we calculated the variance of accuracy improvements across different action categories. The results are shown in the Table 11. Only four methods achieved consistent performance improvements across all action types. Among them, CoME demonstrated the highest relative improvement (+11.56%) while maintaining a lower improvement bias (4.41). Furthermore, we observed that methods incorporating multiple experts, such as MoE and CoME, exhibit lower improvement bias across different action categories compared to Dense models. This indicates that leveraging specialized experts not only enhances overall performance but also contributes to greater robustness of the method.

*Table 11.* Detailed results on AITZ with relative improvement and improvement variance.

| Method | SCROLL | CLICK | TYPE | PRESS | STOP | OVERALL | IMPROVE |
|---|---|---|---|---|---|---|---|
| Average | 43.57 | 51.92 | 74.15 | 57.74 | 69.07 | 55.41 | - |
| AutoUI | $61.40_{+17.83}$ | $32.20_{-19.72}$ | $81.40_{+7.25}$ | $57.70_{-0.04}$ | $74.40_{+5.33}$ | 47.69 | $-7.73_{12.37}$ |
| Qwen2VL(2B) | $23.05_{-20.52}$ | $43.74_{-8.18}$ | $59.40_{-14.75}$ | $52.22_{-5.52}$ | $47.02_{-22.05}$ | 43.80 | $-11.62_{6.53}$ |
| ShowUI | $23.79_{-19.78}$ | $40.34_{-11.58}$ | $85.20_{+11.05}$ | $59.00_{+1.26}$ | $87.55_{+18.48}$ | 49.55 | $-5.87_{14.07}$ |
| UITars(2B) | $30.17_{-13.40}$ | $55.25_{+3.33}$ | $84.77_{+10.62}$ | $53.78_{-3.96}$ | $61.39_{-7.68}$ | 55.63 | $+0.21_{8.40}$ |
| SeeClick | $11.14_{-32.43}$ | $52.96_{+1.04}$ | $53.00_{-21.15}$ | $67.88_{+10.14}$ | $55.36_{-13.71}$ | 49.11 | $-6.31_{15.23}$ |
| Qwen2VL(7B) | $43.28_{-0.29}$ | $55.30_{+3.38}$ | $51.40_{-22.75}$ | $56.92_{-0.82}$ | $64.87_{-4.20}$ | 54.46 | $-0.96_{9.22}$ |
| UIGround | $58.22_{+14.65}$ | $58.48_{+6.56}$ | $73.85_{-0.30}$ | $58.22_{+0.48}$ | $68.78_{-0.29}$ | 60.19 | $+4.77_{5.81}$ |
| OS-Atlas | $76.12_{+32.55}$ | $54.83_{+2.91}$ | $88.60_{+14.45}$ | $68.67_{+10.93}$ | $81.75_{+12.68}$ | 65.11 | $+9.69_{9.75}$ |
| UITars(7B) | $56.50_{+12.93}$ | $63.87_{+11.95}$ | $85.77_{+11.62}$ | $58.22_{+0.48}$ | $71.29_{+2.22}$ | 65.41 | $+9.99_{5.34}$ |
| MolmoE | $28.19_{-15.38}$ | $33.17_{-18.75}$ | $62.00_{-12.15}$ | $37.07_{-20.67}$ | $54.96_{-14.11}$ | 38.23 | $-17.19_{3.95}$ |
| Qwen2VL-MoE | $48.75_{+5.18}$ | $59.43_{+7.51}$ | $74.00_{-0.15}$ | $57.70_{-0.04}$ | $70.83_{+1.76}$ | 60.69 | $+5.27_{3.02}$ |
| AriaUI | $53.73_{+10.16}$ | $60.20_{+8.28}$ | $80.80_{+6.65}$ | $63.70_{+5.96}$ | $76.38_{+7.31}$ | 63.56 | $+8.14_{1.46}$ |
| CoME | $52.07_{+8.50}$ | $65.22_{+13.30}$ | $83.80_{+9.65}$ | $59.53_{+1.79}$ | $83.33_{+14.26}$ | 66.98 | $+11.56_{4.41}$ |

## F. Analysis of CoME architecture flexibility

CoME decouples diverse capabilities by assigning them to different experts, and integrates these capabilities during reasoning via output-oriented activation. CoME can flexibly adjust its architecture according to the capabilities required by a task, and can also scale up the model by increasing the number of experts for each capability. To validate the flexibility of the CoME architecture, we remove the screen-summary expert to create a 3-expert variant and evaluate performance under fewer reasoning stages. To further study scalability, we scale the number of experts to 2x at each stage, resulting in an

8-expert variant. We compare variants with different numbers of experts during the CoT-SFT stage on AITZ dataset, and report the results in the Table 12. The results demonstrate that CoME supports architectures with different numbers of experts, and that action accuracy improves as the number of experts increases.

*Table 12.* Comparison of CoME variants with different numbers of experts.

| Method | #Expert | SCROLL | CLICK | | TYPE | | PRESS | STOP | Overall | |
|---|---|---|---|---|---|---|---|---|---|---|
| | | | type | match | type | match | | | type | match |
| | 3-expert | 41.22 | 81.44 | 61.52 | 83.40 | 79.80 | 66.57 | 71.31 | 74.42 | 62.28 |
| CoME | 4-expert | 52.31 | 81.90 | 62.42 | 75.25 | 71.80 | 63.37 | 69.24 | 75.10 | 62.93 |
| | 8-expert | 54.39 | 81.12 | 61.96 | 83.60 | 79.20 | 56.92 | 73.02 | 75.66 | 63.59 |

## G. Analysis of DPO data selection strategies

For each action, we sample ten reasoning trajectories and construct DPO pairs using one of three strategies as described in Appendix B: (1) **cc**: pair two correct trajectories, if the sampled trajectories are all correct; (2) **cw**: pair a correct trajectory with a wrong one, if the sampled trajectories are partially correct; and (3) **lw**: pair the label with an wrong trajectory, if the sampled trajectories are all wrong. We provide more detailed analysis of information gain comparison among different combination of the pairing strategy as shown in Table 13. Higher InfoGain and InfoPositive will lead to higher action accuracy, because the higher InfoGain indicates that the reasoning trajectory provides more useful information, and higher InfoPositive shows that the intermediate reasoning step is much more valid. Introducing pair data from lw will decrease the InfoGain in reject output, because lw shows that the task is difficult for CoME to finish, the trajectory with the lowest $\mathcal{R}_{IG}$ contains less useful information. Introducing pair data from cc increase the InfoGain in reject output, because some suboptimal correct trajectories are added to the rejected output.

*Table 13.* Analysis of data selection strategy

| Strategy | IG@SS | | IG@AP | | IG@AD | | InfoGain | | InfoPositive | | Action Accuracy | |
|---|---|---|---|---|---|---|---|---|---|---|---|---|
| | reject | choose | reject | choose | reject | choose | reject | choose | reject | choose | type | match |
| **AITZ** | | | | | | | | | | | | |
| cc+cw+lw | 0.0748 | 0.0234 | -0.1430 | 0.1595 | -0.4354 | 0.0814 | -0.5037 | 0.2645 | 0.0291 | 0.6115 | 77.60 | 65.96 |
| cc+cw | 0.0762 | **0.0908** | -0.1294 | 0.1125 | -0.4252 | 0.0689 | -0.4784 | **0.2723** | 0.0300 | **0.7045** | 78.60 | 66.98 |
| cw+lw | 0.0686 | -0.0103 | -0.2049 | **0.1790** | -0.5758 | **0.0905** | -0.7121 | 0.2592 | 0.0380 | 0.4631 | 76.88 | 64.07 |
| cw | 0.0695 | 0.0808 | -0.1973 | 0.1143 | -0.5869 | 0.0741 | -0.7147 | 0.2693 | 0.0409 | 0.5654 | 77.75 | 65.93 |
| **AMEX** | | | | | | | | | | | | |
| cc+cw+lw | 0.0178 | 0.0368 | -0.1735 | **0.1373** | -0.3803 | 0.0327 | -0.5360 | 0.2068 | 0.0159 | 0.4824 | 83.79 | 70.04 |
| cc+cw | 0.0175 | **0.0385** | -0.1629 | 0.1359 | -0.3630 | 0.0337 | -0.5085 | **0.2083** | 0.0172 | **0.5111** | 84.46 | 72.61 |
| cw+lw | 0.0054 | 0.0172 | -0.2268 | 0.1293 | -0.4257 | 0.0342 | -0.6470 | 0.1808 | 0.0183 | 0.3797 | 84.44 | 71.14 |
| cw | 0.0038 | 0.0166 | -0.2199 | 0.1266 | -0.4103 | **0.0357** | -0.6265 | 0.1790 | 0.0199 | 0.4004 | 84.92 | 72.15 |

## H. Analysis of reward strategies

We provide detailed analysis of different strategies in Info-DPO, including different reward model scales and reward function design in Table 14. As for the reward model scale, using 7B reward model can achieve comparable and even better overall performance compared with 72B reward model. While 72B reward model performs much better on CLICK action, because click action is much more challenging that requires precise description of the element to click and coordinates of the element grounding. However, 72B reward results in lower InfoGain which is highly relevant to the action accuracy, because 72B reward model have a better fitness and lower cross-entropy loss, thus the difference of information entropy between different stage is much more smaller. As for different reward function designs, the continuous reward function proves to be more effective, particularly on the CLICK action. This is because a smaller distance between the predicted and ground-truth coordinates indicates higher prediction accuracy, and such outputs should be more likely selected as the chosen output for

model optimization. However, this level of granularity cannot be captured by discrete reward functions.

*Table 14.* Analysis of reward strategy

| Method | ACTION ACCURACY | | | | | | REWARD MARGIN | | | |
|---|---|---|---|---|---|---|---|---|---|---|
| | SCROLL | CLICK | TYPE | PRESS | STOP | OVERALL | IG@SS | IG@AP | IG@AD | InfoGain |
| Reward Model Scale | | | | | | | | | | |
| Reward-7B | 52.07 | 65.22 | 83.80 | 59.53 | 83.33 | 66.98 | 0.0113 | 0.3117 | 0.6610 | 0.9841 |
| Reward-72B | 53.06 | 67.37 | 82.20 | 54.57 | 74.60 | 66.85 | 0.0087 | 0.1681 | 0.6597 | 0.8365 |
| Reward Function Design | | | | | | | | | | |
| Continuous | 52.07 | 65.22 | 83.80 | 59.53 | 83.33 | 66.98 | 0.0113 | 0.3117 | 0.6610 | 0.9841 |
| Discrete | 54.71 | 62.73 | 83.40 | 61.36 | 78.05 | 65.51 | 0.0143 | 0.2379 | 0.5018 | 0.7541 |

# I. Analysis of training reward model

InfoGain reward is estimated based on the difference in information entropy before and after introducing a specific reasoning stage. This estimation leverages a capability that language models naturally possess. Ideally, any logically consistent reasoning should make it easier for a model to predict the correct action. Therefore, we compare the effectiveness of training-based and training-free reward models, as shown in the Table 15. Training-free reward model could achieve comparable performance with training-based one, indicating that the estimation of InfoGain reward is a inner capability of the LLM with the strong general language modeling ability. Experimental results also demonstrate that InfoGain reward is a robust method to evaluate the CoT reasoning process.

*Table 15.* Comparison of training reward model

| Reward Model | Reward Acc | Action Acc |
|---|---|---|
| training-free | 82.24 | 66.64 |
| training-based | 84.68 | 66.98 |

# J. Analysis of efficiency

### J.1. Analysis of GPU memory consumption

In this section, we present a more comprehensive analysis of GPU memory consumption during inference. Specifically, we compare the inference computational cost and action accuracy across three representative architectures: a 7B dense model, a 2B×8 MoE model, and our proposed CoME. The results, summarized in Table 16, demonstrate that CoME achieves the best overall trade-off, delivering the highest action accuracy while maintaining the lowest GPU memory footprint. The key memory advantage of CoME lies in its architecture design: the parameter set dominates the overall memory usage, whereas the additional KV cache overhead incurred during inference remains relatively minor. Taken together, these results underscore that CoME not only achieves best accuracy but also remains resource-efficient.

*Table 16.* Detailed analysis of GPU memory usage

| | Model | Extra | Total | Acc |
|---|---|---|---|---|
| Qwen2VL (7B) | 15.48 GB | 1.04 GB | 16.52 GB | 54.46 |
| UITars (7B) | 15.49 GB | 1.43 GB | 16.92 GB | 65.41 |
| MoE (A3B) | 21.34 GB | 0.98 GB | 22.32 GB | 60.69 |
| CoME (5B) | 10.55 GB | 1.14 GB | 11.69 GB | 66.98 |

## J.2. Analysis of computational cost

In this section, we provide a theoretical analysis of the computational cost. We compare the FLOPs of CoME with strong baseline UITars at each layer. We denote $H$ as the hidden size, $L$ as the sequence length, and $I$ as the FFN's intermediate size of CoME. From the model's configuration files, we can find that:

$$H_{\text{UITars}} \approx 2.33H, \quad I_{\text{UITars}} \approx 2.11I$$

In the standard transformer layer, the computation cost of attention layer and FFN layer can be estimated by:

$$\text{Cost}_{\text{Attn}} = 4LH^2 + 2L^2H$$
$$\text{Cost}_{\text{FFN}} = 2LHI$$

Thus, for CoME, we forward the hidden states from different channels in parallel to the attention layer and the expert layer:

$$\text{Cost}_{\text{CoME}} = 4 \times (4LH^2 + 2L^2H) + 4 \times 2LHI$$
$$= 16LH^2 + 8L^2H + 8LHI$$

For UITars:

$$\text{Cost}_{\text{UITars}} = 4 \times (4L(2.33H)^2 + 2L^2(2.33H)) + 4 \times 2L(2.33H)(2.11I)$$
$$= 21.71LH^2 + 4.66L^2H + 9.83LHI$$

This result shows that CoME has lower computational cost than UITars at the FFN layer. While at the attention layer, as the sequence length in our settings is smaller than the hidden size, the first item is dominant and CoME also has lower computational cost at attention layer.

## K. Case Studies

### K.1. Case study of DPO data pair

In Figure 10, we present the InfoGain values at each stage where the correct output is chosen and the wrong output is rejected. It can be observed that the chosen output exhibit InfoGain greater than zero at each stage, indicating that the reasoning in each stage contributes positively to predicting the correct action. For the rejected outputs, however, during the subtask planning stage the model produced an incorrect next action plan, resulting in negative InfoGain. This shows that the reasoning at this stage has a detrimental impact on predicting the correct action. This case illustrates that InfoGain can be used to evaluate the contribution of intermediate reasoning steps to the prediction of the correct action.

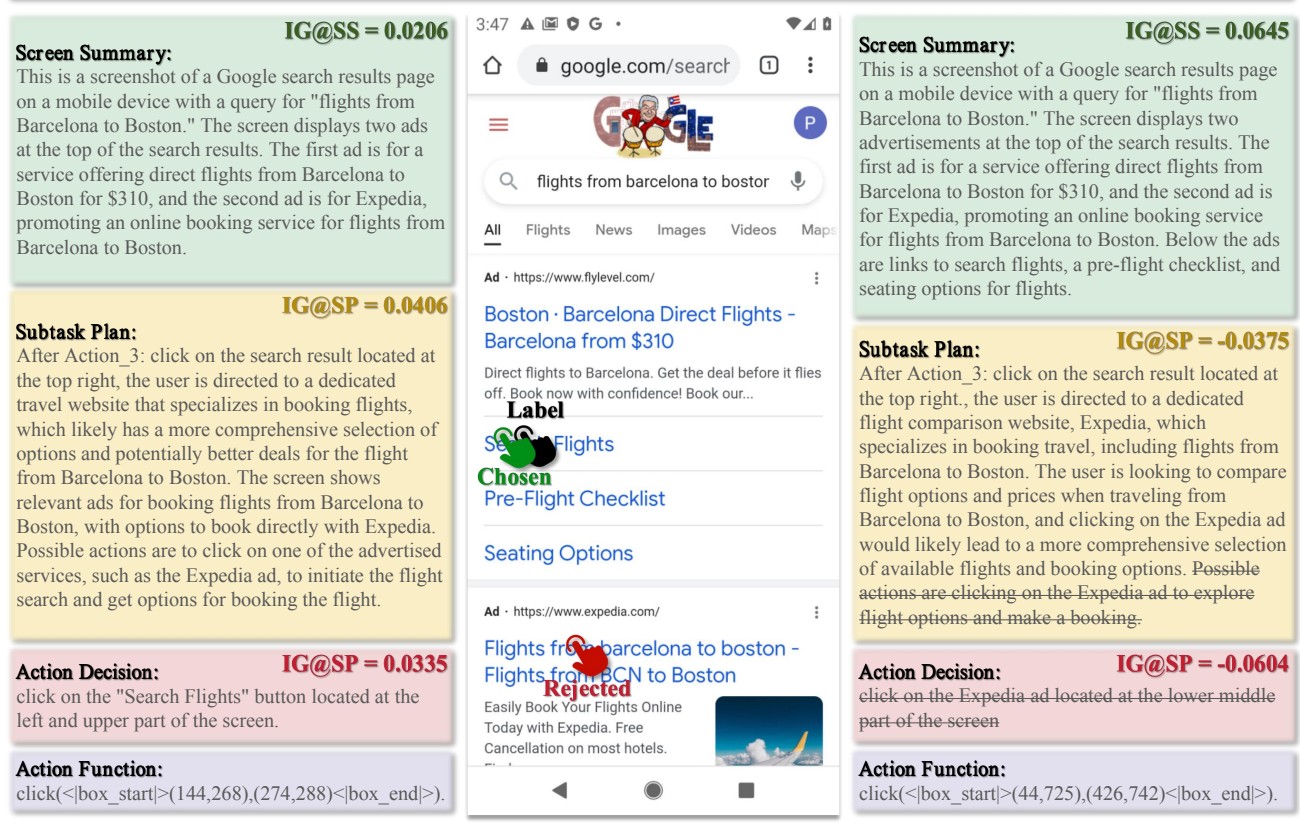

*Figure 10.* InfoGain of DPO data pair where the correct one is chosen and the wrong one is rejected.

In Figure 11, we present the InfoGain values at each stage where both the chosen and rejected outputs are correct. Although the rejected output also result in the correct action, it encounters some reasoning mistakes in subtask plan and action decision stage. Thus the InfoGain at these two stages are negative, indicating that the reasoning process in that stage may be lead to the wrong action. This case illustrates that InfoGain can effectively identify the incorrect intermediate reasoning steps thus suppressing these ineffective reasoning pattern. While this is impossible for the outcome-level reward used by naive DPO.

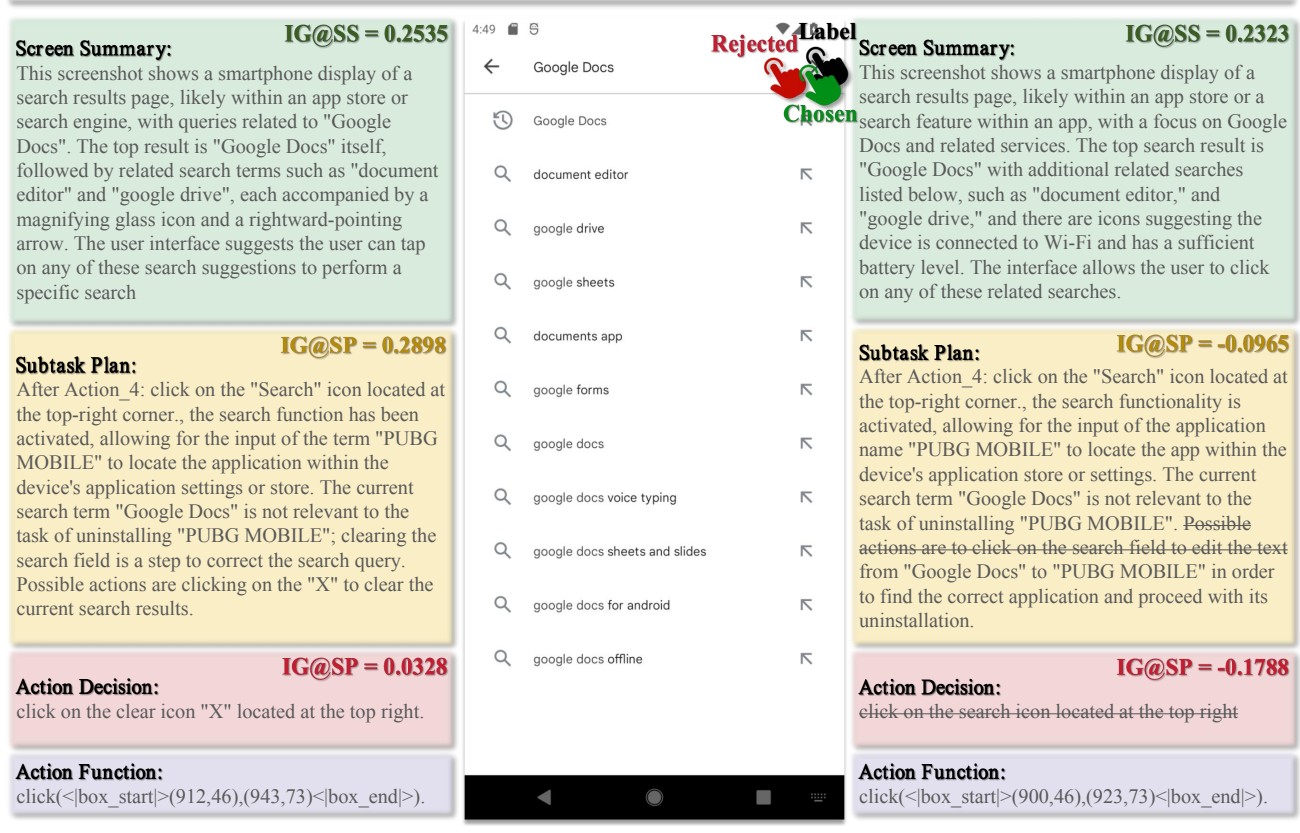

*Figure 11.* InfoGain of DPO data pair where both the chosen and rejected outputs are correct.

