# OpenReview forum: "CoME: Empowering Channel-of-Mobile-Experts with Informative Hybrid-Capabilities Reasoning"
_ICML.cc/2026/Conference — ICML 2026 regular_

### Official Review · Reviewer_LMuo · 2026-03-07

**Soundness:** 2
**Presentation:** 3
**Significance:** 2
**Originality:** 2
**Overall Recommendation:** 4
**Confidence:** 4

**Summary:**

Proposes a mobile GUI agent with stage-specialized experts, activated by output position rather than input token. Adds progressive training + an information-gain-based DPO variant.

**Compliance With Llm Reviewing Policy:**

Affirmed.

**Final Justification:**

It's a reasonable paper with clear score upper board to me. In the best case it will only be interesting to a specific group of readers at most. Therefore, I didnt increase my score.

**Key Questions For Authors:**

no

**Strengths And Weaknesses:**

**Pros**
- Using reward models to estimate entropy reduction per reasoning stage as a process-level reward signal is practically useful and interesting. It yields the largest ablation gain (+4.68%). Cleaner than naive outcome-level DPO for multi-step tasks.

- Results are Good, analysis is papernot bad but also not surprising.

**Cons**

- Output-oriented activation is basically dense soft-MoE. All 4 experts run on every token; the channel router does a softmax blend. This is strictly more expensive than sparse MoE and closely resembles Soft-MoE (Puigcerver et al., 2023) applied to a structured task. Framing it as a new paradigm is a stretch.
- The core of Info-DPO is Ton et al. (ICML 2025) Eq. 4, which CoME directly cites. That's one engineering decision. Calling it a named method ("Info-DPO") and giving it a full bullet point as part of the contribution is probably too much.
- Minor: Margins are relatively small vs. strongest baselines. +1.57% over UITars-7B on AITZ.

Ton et al., Understanding Chain-of-Thought in LLMs through Information Theory




Overall, I think it's still a positive paper but I wont be disappointed if it gets rejected.

---

> ### Author Rebuttal · Authors · 2026-03-30
>
> We sincerely appreciate your effort in reviewing our work and constructive suggestions. We will response to each of your questions in detail; if your concerns are resolved, we would be grateful for your continued support of our work.
>
> ---
>
> > Con1:
>
> We compared the inference memory consumption. The results show that, CoME requires only about half the inference memory of MoE-A3B under the current relatively short-context setting, while still achieving better performance.
>
> ||Model|Extra|Total|Acc|
> |-|-|-|-|-|
> |MoE(A3B)|21.34GB|0.98GB|22.32GB|60.69|
> |CoME(5B)|10.55GB|1.14GB|11.69GB|66.98|
>
> We agree that CoME shares some similarity with soft-MoE, in the sense that both avoid hard Top-K routing and instead use a fully differentiable softmax-based mechanism for expert selection. However, we would like to respectfully clarify that there is still a fundamental difference between soft-MoE and CoME in terms of the activation paradigm.
>
> More specifically, soft-MoE still follows the conventional input-oriented activation paradigm, where experts are selected to process input tokens. As discussed in Appendix A, this design is not well suited for hybrid-capability reasoning, since different reasoning stages often require different capabilities. Ideally, the token generated at each stage should come from the expert best matched to that stage, and the preceding hidden states used for generation should also be produced by the same expert. This is exactly the output-oriented activation paradigm adopted by CoME. Such behavior cannot be achieved in standard MoE, because expert assignments are fixed once earlier tokens are processed. As a result, later reasoning tokens may depend on hidden states that were not produced by the same expert, creating a misalignment between expert activation and the actual reasoning stage. From this perspective, CoME can be viewed as a distinct paradigm.
>
> We believe this distinction is particularly important for multi-stage hybrid-capability reasoning, since **CoME is able to align expert activation with the capability required at each reasoning stage**, thereby allowing the corresponding expert to generate the token for that stage more appropriately.
>
> ---
>
> > Con2: Contribution of Info-DPO
>
> **We fully acknowledge the importance of the work by Ton et al.**, and we have provided explicit citation to it in the paper. However, we would like to respectfully clarify the distinction between their work and ours.
>
> More specifically, Ton et al. show that InfoGain can be used to assess the effectiveness of intermediate reasoning steps. This provides an important theoretical foundation, but it does not study how to use InfoGain as an effective training signal. **Our work builds on this foundation and takes a further step by investigating how to leverage InfoGain for more effective DPO training.**
>
> To this end, we conduct extensive experiments to analyze how the reward signal derived from InfoGain can be used to construct effective DPO training pairs. Our empirical results in Figure-6&Table-6&Table-11 further show that simply using InfoGain to filter DPO pairs is not sufficient. Instead, **effective data construction also requires considering both the magnitude of each step’s contribution and the InfoGain gap between positive and negative samples**. We therefore believe that Info-DPO constitutes a meaningful contribution of our work.
>
> In the revised version, **we will place greater emphasis on how InfoGain is used to enable effective DPO training**, rather than framing our contribution as proposing InfoGain itself for effectiveness evaluation.
>
> ---
>
> > Con3: Margins are relatively small vs. strongest baselines
>
> Compared with OS-Atlas and UI-Tars, **CoME performs better with both less training data and activated parameters**. Statistical significance tests show that CoME achieves 66.89±0.14@AITZ and 72.64±0.16@AMEX, significantly outperforming the strongest baseline (p < 0.01). Importantly, CoME uses a 5B model, whereas OS-Atlas and UI-Tars are both based on 7B models. In addition, CoME uses only about 13K screenshots , which is far smaller than the **million-scale** data used by these baselines. We believe this is particularly meaningful in the mobile setting, where large-scale data collection is costly and often constrained by resource and risk-control challenges. The main goal of CoME is to improve data efficiency and achieve stronger performance with lower data requirements. We believe this result is particularly meaningful in the mobile setting, where large-scale data collection is highly costly and often accompanied by substantial challenges in resource consumption and risk control.
>
> ||#Param|#Data|AITZ|AMEX|
> |-|-|-|-|-|
> |OS-Atlas |7B|2.3M screens, 13.6M elements|65.11|69.99|
> |UI-Tars|7B|6M tutorials, 50B trained tokens|65.41|70.33|
> |CoME|5B|~10k screens|66.98|72.61|

---

> > ### Author Rebuttal · Reviewer_LMuo · 2026-04-01
> >
> > Thank you for the clarifications. I am maintaining my score and removing "Overall, I think it's still a positive paper but I wont be disappointed if it gets rejected."

---

> > > ### Author Response · Authors · 2026-04-02
> > >
> > > We sincerely appreciate your thorough review of our work and the insightful suggestions you provided. We are pleased that our responses have successfully addressed your concerns. If you have any other questions, we are more than willing to have a more in-depth discussion with you during the discussion stage to fully address your concerns. We sincerely appreciate for your support, and we look forward to your continued support in the next stages. Thank your once again for your kindly engagement and timely response.

---

### Official Review · Reviewer_piPM · 2026-03-11

**Soundness:** 3
**Presentation:** 3
**Significance:** 3
**Originality:** 3
**Overall Recommendation:** 5
**Confidence:** 4

**Summary:**

The paper presents a new model architecture and corresponding training methods for mobile GUI agents, called Channel-of-Mobile-Experts (CoME). Based on the observation that mobile agents employ multi-stage reasoning for GUI operations, the paper proposes to integrate specialized experts into the model architecture to explicitly handle each reasoning stage (screen understanding, planning, decision, and action). The authors also introduce a progressive training strategy and an InfoGain-driven DPO method to mitigate error propagation across stages. CoME demonstrates promising improvements on the AITZ and AMEX benchmarks and is more memory-efficient than several baselines at comparable accuracy.

**Compliance With Llm Reviewing Policy:**

Affirmed.

**Final Justification:**

During the rebuttal, authors provides clarifications and new exps results. Especially the results on OOD datasets, and on AndroidControl.

Altough the results on AndroidWorld is still missing, I understand it may require more engineering efforts (while results on AndroidControl is convincing).

I will raise my score from 4 to 5.

**Key Questions For Authors:**

- One major concern is that the CoME design may lack flexibility, as the introduced experts are closely coupled with empirically designed reasoning stages. To address this issue, the following two sub-questions can be considered:
1) To demonstrate that the leveraged reasoning stages are generalizable across mobile GUI tasks. Since this four-stage reasoning process is used in the AITZ datasets, it would be worthwhile to investigate whether the trained CoME (with 4 experts) can be applied to other mobile GUI benchmarks, such as Android World.
2) To decouple the number of experts from the number of reasoning stages. For example, CoME could be evaluated with 3 or 8 experts as shown in Table 10. This approach raises several questions: a) Does the performance improvement stem solely from overparameterization? b) What would happen if a conventional MoE training strategy is applied? For instance, it would be appropriate to include Qwen2VL-MoE-5B as a baseline. The current comparison with Qwen2VL-MoE-3B is not fair due to the difference in model sizes.

- The current evaluation only demonstrates in-domain performance, i.e., models are trained and tested on the same datasets AITZ and AMEX. Additional out-of-domain results would be highly beneficial.

**Limitations:**

yes

**Strengths And Weaknesses:**

Strengths:
- The paper is well motivated and easy to follow.
- The proposed methods are clearly presented and thoroughly validated through ablation studies.
- The evaluation results are promising compared to a range of baselines.

Weaknesses:
- The architecture is closely coupled with the fixed reasoning stages.
- CoME is trained on AITZ and AMEX, so the comparison with other baselines on these datasets may not be entirely fair.

---

> ### Author Rebuttal · Authors · 2026-03-30
>
> We sincerely thank you for your insightful suggestions and valuable comments! We will try to address your concerns in detail, and we would appreciate it very much if you could kindly raise your score if your concerns are addressed.
>
> ---
>
> > W1: The architecture's flexibility.
>
> Thanks for your valued question. **Actually, the architecture of CoME is not fixed; rather, it can be flexibly adjusted according to the underlying reasoning stages**. Our framework also supports variants with different numbers of experts. For example, in Appendix F, we provide a 3-expert variant of CoME in which the screen summary stage is removed. The resulting performance drop suggests that the screen summary stage is indeed important for effective reasoning. In addition, we can also allocate multiple experts to each reasoning stage. We also present an 8-expert variant in Appendix F, where each reasoning stage is associated with two experts. The results show that scaling the number of experts can further improve performance. More broadly, the flexibility of CoME lies in its extensibility: if a new reasoning stage is needed, one can train a corresponding expert using data from that stage, insert the expert into CoME, and then retrain the router to incorporate the new capability.
>
> ||Type|Match|
> |-|-|-|
> |3-expert|74.42|62.28|
> |4-expert|75.10|62.93|
> |8-expert|75.66|63.59|
>
> CoME can also accommodate flexible reasoning stages. As shown in Table 4, stages such as caption, OCR, and grounding each only rely on a specific capability and may correspond to only a particular part of the reasoning process. Accordingly, CoME can activate only the relevant expert when needed during inference. Therefore, CoME is not strictly tied to a fixed reasoning-stage decomposition; instead, **CoME can flexibly select which experts to activate according to the specific needs of the reasoning process**.
>
> ---
>
> > W2: Fair comparison
>
> In fact, **we also fine-tuned the baselines on the AITZ and AMEX datasets**, while ensuring that the data format remained aligned with each model’s original training format so as to avoid affecting their native capabilities. Moreover, baselines such as UI-Tars and OS-Atlas were pretrained with substantially more data than CoME, which puts them at an advantage from the data perspective. Nevertheless, their performance still remains below that of CoME. For a more strictly controlled comparison, **the Qwen-series baselines provide the fairest setting**, and the results under this setting are sufficient to demonstrate the effectiveness of CoME.
>
> ||#Param|#Data|AITZ|AMEX|
> |-|-|-|-|-|
> |OS-Atlas |7B|2.3M screens, 13.6M elements|65.11|69.99|
> |UI-Tars|7B|6M tutorials, 50B trained tokens|65.41|70.33|
> |CoME|5B|~10k screens|66.98|72.61|
>
> ---
>
> > Q1.1: Evaluation on AndroidWorld
>
> Thank you very much for this constructive suggestion. We have evaluated the effectiveness of CoME on the AndroidWorld benchmark. All of the methods are trained on AITZ and the results indicate that CoME remains effective in this setting as well. We will consider incorporating a more comprehensive experimental analysis into the revised version.
>
> ||SR|
> |-|-|
> |Qwen2VL-7B|12.70|
> |OS-Atlas-7B|15.87|
> |MoE-A5B|9.52|
> |CoME-5B|23.80|
>
> ---
>
> > Q1.2: Comparison with MoE-A5B
>
> Table 10 shows that increasing the number of experts improves performance, suggesting that CoME can scale effectively. Moreover, MoE-A3B has a total of 11B parameters, already exceeding the 5B parameters of CoME. We implemente MoE-A5B. The results show that CoME still outperforms the MoE model even when both activate approximately 5B parameters. This further suggests that, **under the same activated parameter budget, CoME is able to achieve better reasoning performance** through more effective expert capability decoupling and better alignment between expert activation and the reasoning process.
>
> ||Type|Match|
> |-|-|-|
> |A3B|72.94|60.69|
> |A5B|75.06|60.94|
> |CoME|75.10|62.93|
>
> We train CoME using a conventional MoE training strategy. The results show that such a strategy is unable to reliably activate the appropriate expert at the corresponding reasoning stage. This misalignment further leads to a decline in action prediction accuracy. These results suggest that **aligning expert activation with the reasoning stages is important, and that the proposed training strategy is necessary**.
>
> ||Type|Match|Align|
> |-|-|-|-|
> |MoE strategy|74.87|61.32|30.49|
> |CoME strategy|75.10|62.93|98.81|
> ---
>
> > Q2: OOD results
>
> We further trained CoME on the AndroidControl and evaluated it on the OOD split. The results show that CoME outperforms existing methods under OOD settings. In addition, its zero-shot performance on AMEX provides further evidence that **CoME achieves meaningful OOD generalization capability**.
>
> ||app_un|task_un|category_un|AMEX|
> |-|-|-|-|-|
> |Qwen2VL2B|51.25|51.94|50.60|36.34|
> |UITars2B|58.90|60.21|58.59|46.41|
> |Qwen2VL7B|63.19|64.65|63.68|46.64|
> |OSAtlas7B|63.56|64.98|63.58|51.68|
> |CoME|65.29|67.23|65.84|55.02|

---

> > ### Author Rebuttal · Reviewer_piPM · 2026-04-01
> >
> > Thanks for the efforts and reply. I think these results have addressed some of my questions.
> >
> > One remaining concern is the relatively low performance of CoME on AndroidWorld (23.80%). Is this because CoME is trained only on AITZ and AMEX and relies on four-stage reasoning steps, which may not be well suited to AndroidWorld tasks?
> >
> > To better demonstrate the effectiveness and flexibility of CoME, would it be possible to fine-tune CoME on AndroidWorld with task-specific reasoning stages (and experts)? Achieving stronger performance on AndroidWorld could make the results more convincing.

---

> > > ### Author Response · Authors · 2026-04-01
> > >
> > > Thank you very much for your effort and timely reply, and we sincerely appreciate your acknowledgment that our rebuttal has addressed most of your concerns. We are also very grateful for the opportunity to further clarify the remaining questions.
> > >
> > > ---
> > >
> > > First, we would like to clarify that the AITZ training data mainly consists of relatively simple tasks involving web browsing and Google system apps, with an average trajectory length of 7.5 steps. In contrast, AndroidWorld primarily involves third-party apps and much longer trajectories (14.2 steps on average), making the tasks substantially more challenging. This creates a notable distribution gap between the training and evaluation settings, making the training data is not well suited to AndroidWorld tasks. Under such a challenging setting, CoME still outperforms the baselines trained on the same data by a clear margin (+8.07). We believe this provides supporting evidence for the effectiveness of CoME under substantial distribution shift.
> > >
> > > In addition, as AndroidWorld trajectories are relatively long, failure at an intermediate step often causes the entire task to fail. As a result, the currently reported end-to-end task success rate may be too coarse-grained as an evaluation metric. To provide a more informative assessment, we additionally report a relative completion metric, defined as the proportion of the trajectory prefix that is completed correctly. Under this metric, CoME again achieves the best performance among methods trained on the same data, with a margin of +9.34, which further supports the effectiveness of our method.
> > >
> > > | Method     | E2E-SR | Rel-SR |
> > > |------------|-------:|-------:|
> > > | Qwen2VL-7B | 12.70  | 37.18  |
> > > | OS-Atlas-7B| 15.87  | 39.18  |
> > > | MoE-A5B    | 9.52   | 27.19  |
> > > | CoME-5B    | 23.80  | 48.53  |
> > >
> > > ---
> > >
> > > Regarding the possibility of fine-tuning on AndroidWorld, our current limitation is that AndroidWorld is released only as a benchmark, and the official benchmark does not provide training data. We have made our best effort to search for publicly available AndroidWorld training data, but unfortunately were unable to find a usable open-source training set. As a result, we are not yet able to complete fine-tuning experiments on AndroidWorld in such short time. To better address your concerns, we have already begun constructing training data and collecting trajectories for AndroidWorld by ourselves, and we will complete the training and evaluation as soon as possible. Due to the very limited time, we were only able to collect and filter 200 valid training trajectories, while ensuring that they had no overlap with the test set. Even under this limited setting, the results show that fine-tuning on in-domain AndroidWorld data leads to clear performance gains (+10.40), and CoME still outperforms the baseline methods (+12.02). We believe that collecting more training data would likely yield further improvements, and we will continue to expand and enrich the training set.
> > >
> > > | Method     | AW |
> > > |------------|-------|
> > > | Qwen2VL-7B | 20.63  |
> > > | OS-Atlas-7B | 22.22 |
> > > | MoE-A5B    | 17.46   |
> > > | CoME-5B    | 34.20  |
> > >
> > > ---
> > >
> > > Regarding the effectiveness and flexibility of CoME, we believe our response to Q2 also provides relevant supporting evidence. We trained CoME on AndroidControl and evaluated it on the corresponding OOD split. The results show that, under the same training data and experimental settings, CoME still outperforms the baselines. We believe these findings further support the effectiveness and flexibility of CoME.
> > >
> > > | Method    | app_un | task_un | category_un |
> > > |-----------|-------:|--------:|------------:|
> > > | Qwen2VL2B | 51.25  | 51.94   | 50.60       |
> > > | UITars2B  | 58.90  | 60.21   | 58.59       |
> > > | Qwen2VL7B | 63.19  | 64.65   | 63.68       |
> > > | OSAtlas7B | 63.56  | 64.98   | 63.58       |
> > > | CoME      | 65.29  | 67.23   | 65.84       |
> > >
> > > ---
> > >
> > > We hope that the additional results and clarifications we have provided help address your concern. We also sincerely hope for your understanding that, due to the lack of publicly available training data and the limited time available during the rebuttal period, we are unable to complete fine-tuning experiments on AndroidWorld at this stage. But we promise that we will make our best effort to complete these experiments as soon as possible and provide the corresponding results before the end of the discussion stage. If you feel that your concern has been addressed, we would be truly grateful if you would kindly consider raising your score. Thank you again for your time, effort, and thoughtful feedback.

---

### Official Review · Reviewer_rJai · 2026-03-12

**Soundness:** 3
**Presentation:** 3
**Significance:** 3
**Originality:** 3
**Overall Recommendation:** 3
**Confidence:** 4

**Summary:**

This paper proposes CoME (Channel-of-Mobile-Experts), a novel architecture for mobile agents powered by multimodal large language models (MLLMs). The work addresses a key challenge in mobile agent systems: hybrid-capabilities reasoning, which involves several sequential abilities required to complete mobile UI tasks: Screen understanding, subtask planning, taking action. The authors argue that existing mobile agents struggle to both specialize and integrate these capabilities effectively. To address this, the paper introduces: CoME, where different experts correspond to distinct reasoning and skill specific expert. Instead of standard mixture-of-experts routing, the framework introduces output-oriented activation, where the expert responsible for a reasoning stage generates the output tokens. To Train CoME authors propose a multi-stage training strategy which first: does Expert-FT – capability-specific finetuning of experts using stage-specific data, next Router-FT – training a routing mechanism to activate the correct expert at each reasoning stage. Followed by CoT-FT – chain-of-thought finetuning for coherent multi-step reasoning and finally Info-DPO – an information gain–based preference optimization to improve reasoning trajectories.

**Compliance With Llm Reviewing Policy:**

Affirmed.

**Final Justification:**

Thank you for the detailed rebuttal and clarifications. I acknowledge the rebuttal from the authors but some of my concerns are still valid so I will keep my original score

**Key Questions For Authors:**

Questions:

1. Given that improvements on the benchmarks with the new architecture is small (~2-3%) I am curious to hear from authors do they believe the trade-offs made by the architecture by introducing a task/skill-specific priors is worth it compare to the cost/complexity associated with multi-stage data curation and training in addition to inference inefficiency at scale? I am a bit on fence about whether such a approach has practical value. It’d be great if authors can add more thoughts and arguments regarding this.

I am happy to increase my rating if authors address my concerns. My main concerns are around practicality of the method

**Limitations:**

Mentioned above

**Strengths And Weaknesses:**

Strengths

1. Overall improvements in performance on both AMEX and AITZ datasets when compared with models at similar or slightly larger scale.
2. The paper is written well and proposed approach is intuitive. It adds a simple inductive bias in the model architecture which can lead to better sample efficiency as model leverages priors about skills required for the task.

Weaknesses:

1. Eventhough the proposed approach is intuitive it seems to be baking task specific priors into model architecture which comes at a cost of:
    1. Complex multi-stage training pipeline which makes already complex pipeline of training multi-modal LLMs hard to manage. Each stage requires curation of very specific kind of data. Scaling such a approach to general tasks seems difficult but is okay when specializing a model for a specific domain.
    2. Stage/Subtask/Skill specific routers make inference inefficient as we need task specific routing during different stages of tasks during inference. Running batch inference for such MoE models in a scalable way is difficult.
2. As far as I understand the paper only presents results on static GUI benchmarks. However it is important to test efficacy of the model on online environments like AndroidWorld, WebArena, or OSWorld (only a subset) that demonstrates the capability of the model to complete GUI use tasks. This will strengthen the experiments section and make paper more convincing.

---

> ### Author Rebuttal · Authors · 2026-03-30
>
> We sincerely appriciate for your constructive feedback and insightful comments. We will address each of your questions in detail, and we would be most grateful if you might consider raising your score once your concerns have been fully resolved.
>
> ---
>
> > W1.1: Complexity and generalization
>
> Our three-stage pipeline follows standard agent training practice: Expert-FT corresponds to continue pre-training, CoT-FT to supervised fine-tuning, and the only extra stage, Router-FT, is lightweight and incurs minimal cost. Therefore, **we believe it doesn't increases pipeline complexity**. In addition, our method only requires annotated multi-stage CoT data, which can be directly split by stage to train different experts, without any extra expert-specific annotation.
>
> CoME and the proposed multi-stage training strategy are primarily designed for agent tasks, which often follow the ReAct paradigm and thus naturally fit stage-wise hybrid-capability reasoning. More broadly, CoME can also be applied to other tasks with similar staged reasoning structures. We train CoME with LLaVA-CoT-100K and results on ScienceQA further suggest that **CoME generalizes beyond agent tasks to more general visual reasoning settings.**
>
> | | SQA|
> |-|-|
> |Qwen2VL|81.90|
> |MoE|83.39|
> |CoME|85.62|
> ---
>
> > W1.2: Batch inference efficiency
>
> We compare the inference efficiency of CoME against MoE-A5B, and reported the throughput (tokens/second) for processing 96 samples under different batch sizes. The results show that the inference speed of CoME is comparable to that of MoE, with MoE achieving roughly 1.0–1.25× the throughput of CoME. MoE encounters OOM once the batch size exceeds 16, whereas CoME can scale to a batch size of 24. Therefore, **although CoME is slightly slower, it can support larger batch sizes under the same memory budget**. As a result, the maximum throughput under an equal memory budget is comparable.
>
> ||1|8|16|24|ACC|
> |-|-|-|-|-|-|
> |MoE-A5B|18.1|92.3|178.6|OOM|60.9|
> |CoME-5B|17.1|76.7|142.1|180.0|62.9|
> ---
>
> > W2: Evaluation on AndroidWorld
>
> Thank you very much for this constructive suggestion. We have evaluated the effectiveness of CoME on the AndroidWorld benchmark. All of the methods are trained on AITZ and the results indicate that **CoME remains effective on online benchmark AndroiWorld** as well. We will consider incorporating a more comprehensive experimental analysis into the revised version.
>
> ||SR|
> |-|-|
> |Qwen2VL-7B|12.70|
> |OS-Atlas-7B|15.87|
> |MoE-A5B|9.52|
> |CoME-5B|23.80|
> ---
>
> > Q1: Discussion regarding practical value
>
> Compared with OS-Atlas and UI-Tars, **CoME performs better with both less training data and activated parameters**. Statistical significance tests show that CoME achieves 66.89±0.14@AITZ and 72.64±0.16@AMEX, significantly outperforming the strongest baseline (p < 0.01). Importantly, CoME uses a 5B model, whereas OS-Atlas and UI-Tars are both based on 7B models. In addition, CoME uses only about 13K screenshots , which is far smaller than the **million-scale** data used by these baselines. We believe this is particularly meaningful in the mobile setting, where large-scale data collection is costly and often constrained by resource and risk-control challenges. The main goal of CoME is to improve data efficiency and achieve stronger performance with lower data requirements.
>
> ||#Param|#Data|AITZ|AMEX|
> |-|-|-|-|-|
> |OS-Atlas |7B|2.3M screens, 13.6M elements|65.11|69.99|
> |UI-Tars|7B|6M tutorials, 50B trained tokens|65.41|70.33|
> |CoME|5B|~10k screens|66.98|72.61|
>
> For data construction, **we only require annotated multi-stage CoT data, which can be directly split by stage to train different experts**, without any additional expert-specific annotation. This makes our method annotation-efficient and allows it to better exploit existing data.
>
> For the multi-stage training procedure, **we believe it doesn't substantially increases the pipeline complexity.** In practice, agent training already commonly follows a continued pretraining + post-training pipeline: Expert-FT corresponds to the former, and CoT-FT corresponds to the latter. The only additional stage is a lightweight Router-FT, which uses only around 10K samples to train a single linear layer, and thus introduces very limited extra cost.
>
> For the inference efficiency, **CoME is also comparable to MoE in terms of inference speed**. Moreover, CoME incurs lower memory overhead, which allows the use of larger batch sizes during inference.
>
> Overall, CoME and the proposed multi-stage training strategy are designed primarily for agent tasks, which often follow the ReAct paradigm of Observation–Thought–Action and thus naturally fit stage-wise hybrid-capability reasoning. More broadly, CoME can also be extended to other tasks with similar staged reasoning structures, as supported by the results on ScienceQA. Taken together, these findings suggest that **CoME has practical utility and can generalize to a broader range of staged-reasoning CoT tasks**.

---

> > ### Author Rebuttal · Reviewer_rJai · 2026-04-06
> >
> > NA

---

> > > ### Author Response · Authors · 2026-04-06
> > >
> > > We greatly appreciate the time and effort you have devoted to reviewing our work, as well as your many valuable suggestions, which have helped strengthen the paper. We are very encouraged that our detailed explanations and additional experiments fully address your concerns. We would be very grateful if you could kindly consider raising your score accordingly, as you mentioned may be possible. If you still feel that some concerns remain unresolved, we would greatly appreciate your further feedback and the opportunity to continue the discussion with you.

---

### Official Review · Reviewer_seuz · 2026-03-19

**Soundness:** 3
**Presentation:** 2
**Significance:** 2
**Originality:** 2
**Overall Recommendation:** 4
**Confidence:** 3

**Summary:**

This paper proposes CoME for mobile agents that addresses hybrid-capabilities reasoning through four stage-specific experts (screen summary, subtask planning, action decision, action function). The key contributions include: (1) output-oriented expert activation mechanism, (2) progressive training strategy (Expert-FT, Router-FT, CoT-FT), and (3) InfoGain-Driven DPO (Info-DPO) that evaluates intermediate reasoning steps via information gain. Experiments on AITZ and AMEX datasets demonstrate state-of-the-art performance with 66.98% and 72.61% overall accuracy respectively.

**Compliance With Llm Reviewing Policy:**

Affirmed.

**Final Justification:**

Thank you for the response. The author solved most of my questions.

**Key Questions For Authors:**

1.The concept of using information gain to evaluate reasoning steps has been explored in prior work on process reward modeling. The paper does not clearly distinguish Info-DPO from existing approaches like PRM or step-level evaluation methods.

2.Although Table 8 reports GPU memory and inference time, the paper does not discuss the training cost of the three-stage progressive training strategy.

**Limitations:**

Yes

**Strengths And Weaknesses:**

strengths

1.The paper introduces a well-motivated architecture that explicitly decouples different reasoning capabilities into specialized experts.

2.InfoGain-Driven DPO offers a principled approach to evaluate intermediate reasoning steps beyond outcome-level rewards.

3.Comprehensive experiments on two benchmarks with extensive baselines demonstrate clear performance gains.

weaknesses：

1.The evaluation compares the 5B-parameter CoME against a 3B-parameter Qwen2VL-MoE baseline, introducing a significant 67% parameter discrepancy. Given that the ablation study (Table 3) shows only a 4.6% gain for MoE over the Dense baseline on AITZ, it is highly likely that CoME's improvements stem primarily from increased parameter capacity rather than architectural efficacy.

2.The sota performance is overclaimed and lacks compelling empirical backing. The performance improvements over leading 7B baselines are minimal, yielding only +1.87% over OS-Atlas on AITZ and +2.28% over UITars on AMEX.

3.The evaluated baselines strictly employ Chain-of-Thought (CoT) reasoning. Consequently, the absence of a non-CoT Supervised Fine-Tuning (SFT) baseline leaves it unclear whether the performance improvements are attributable to CoME's architectural innovations, or merely to the inherent benefits of CoT prompting.

4.The proposed 'output-oriented activation' essentially functions as an output-token-based soft routing mechanism. However, the paper does not adequately claim the fundamental differences between this approach and standard MoE architectures, particularly for complex reasoning.

---

> ### Author Rebuttal · Authors · 2026-03-30
>
> We sincerely thank you for your constructive suggestions and valuable comments! We will answer the questions in detail and would appreciate it very much if you could kindly raise your score if your concerns are addressed.
>
> ---
>
> > W1: CoME's benefits from increased parameter
>
> MoE-A3B has approximately 11B total parameters and 3B activated parameters. Following your suggestion, we further implemented MoE-A5B variant. The results show that, although MoE-A5B outperforms MoE-A3B very limited (+0.25%), suggesting that **simply increasing activated parameters does not lead to improvement**. CoME still surpasses MoE-A5B by 2%,demonstrating that the improvement of CoME does not stem merely from parameter scaling. Rather, **it arises from CoME’s more effective expert decoupling, which better aligns expert activation with the reasoning process**.
>
> ||Type|Match|
> |-|-|-|
> |A3B|72.94|60.69|
> |A5B|75.06|60.94|
> |CoME|75.10|62.93|
> ---
>
> > W2: The sota performance is overclaimed
>
> We fully agree that terms such as state-of-the-art (SOTA) should be used with considerable caution. For this reason, we **do not intend to overclaim CoME as the SOTA method**. We therefore believe that the current concern may stem from a possible *misunderstanding*. Compared with OS-Atlas and UI-Tars, **CoME performs better with both less training data and activated parameters.** Statistical significance tests show that CoME achieves 66.89±0.14@AITZ and 72.64±0.16@AMEX, significantly outperforming the strongest baseline (p < 0.01).
> In terms of model scale, CoME uses only 5B parameters, while OS-Atlas and UI-Tars are both 7B models—about 40% larger—yet still underperform CoME.
> In terms of data, CoME uses substantially less training data (only about 13K screenshots), far fewer than the million-scale data used by OS-Atlas and UI-Tars. For fairness, we additionally fine-tuned both baselines on AITZ with data formats aligned to their original setup. Even so, **although these baselines rely on roughly two orders of magnitude more data, CoME still achieves better performance.**
> We believe this result is particularly meaningful where large-scale data collection is costly and constrained by resource and risk-control challenges. The main goal of CoME is to improve data efficiency. We hope that the above clarification and evidence help resolve this your misunderstanding.
>
> ||#Param|#Data|AITZ|AMEX|
> |-|-|-|-|-|
> |OS-Atlas |7B|2.3M screens, 13.6M elements|65.11|69.99|
> |UI-Tars|7B|6M tutorials, 50B trained tokens|65.41|70.33|
> |CoME|5B|~10k screens|66.98|72.61|
>
> ---
> > W3: Lack of non-CoT SFT baseline.
>
> In fact, we have included non-CoT baseline in the experiment (UGround). To address your concern, we add Qwen-based non-CoT baselines. The results show that CoT-based baseline performs better than non-CoT(+4.14), suggesting that CoT reasoning is beneficial. The CoT-based results show that, **CoME achieves better result trained on the same CoT data (+12.52)**.
>
> ||Type|Match|
> |-|-|-|
> |Q2VL(non)|69.82|50.32|
> |Q2VL(CoT)|71.59|54.46|
> |CoME|78.60|66.98|
> ---
>
> > The fundamental differences between CoME and MoE
>
> As noted in Footnote 2 on page 3, in hybrid-capability reasoning, different reasoning stages rely on different abilities. Ideally, we would like different experts to handle the corresponding reasoning steps and generate tokens. Ideally, the preceding tokens' hidden state should likewise be processed by the same expert. This is **output-oriented activation** paradigm. However, in MoE, the expert assignment for earlier tokens is fixed, and cannot be adjusted during the generation. As a result, later reasoning tokens may depend on hidden states that were not produced by the same expert, causing the misalignment between expert activation and reasoning stage. In contrast, CoME uses a channel router to ensure that the hidden states used at each reasoning stage come from the expert specialized for that stage.
>
> ---
> > Q1: Distinguish Info-DPO with PRM
>
> However, traditional PRM would require substantial step-level annotation to explicitly identify which intermediate reasoning steps are incorrect. **Such annotation is highly labor-intensive and expensive**. **In contrast, InfoGain does not require any manual annotation of intermediate steps**. It only relies on the final answer to train the reward model. Appendix I further shows that a training-free reward model achieves comparable results, supporting InfoGain as a low-cost yet effective way to assess intermediate reasoning steps. We also included a comparison with a step-level evaluation method via simulated Monte Carlo tree sampling (MobileIPL). The results show that **Info-DPO is more effective**.
>
> ||Acc|
> |-|-|
> |SFT|62.25|
> |DPO|65.37|
> |MobileIPL|65.36|
> |Info-DPO(free)|66.64|
> |Info-DPO|66.98|
> ---
> > Q2: Training cost
>
> We have further summarized the amount of data used at each training stage, together with the corresponding training time required.
>
> ||Data/Time(min)|
> |-|-|
> |Expert-FT|13k,10|
> |Router-FT|13k,167|
> |CoT-FT|13k,318|

---

> > ### Author Rebuttal · Reviewer_seuz · 2026-04-04
> >
> > Thank you for the response. The author solved most of my questions.

---

> > > ### Author Response · Authors · 2026-04-04
> > >
> > > We sincerely appreciate your time and effort in providing such constructive reviews, as well as your kind recognition of our work. In response to your valuable suggestions, we have performed additional experiments and analyses. We are glad that the supplementary information we provided has resolved your concerns and met with your approval. We promise that these additional analyses will be incorporated into the revised version of the paper. We would be greatly grateful for your continued support of our work.

---

### Decision · Program_Chairs · 2026-04-30

**Decision:**

Accept (regular)

**Comment:**

**Summary:**

The paper proposes CoME, a mobile GUI agent architecture that splits the task into four reasoning stages and assigns a specialized expert to each one. It also introduces a staged training pipeline and an information-gain-based preference optimization method to improve intermediate reasoning quality. Experiments on AITZ and AMEX show strong results, suggesting that this stage-specialized design can improve mobile agent performance.

Strengths and weaknesses from initial reviews:

**Strengths:**

1. The paper is well motivated and based on a clear idea: different stages of mobile-agent reasoning may benefit from different specialized experts.
2. The method is presented clearly, and the staged training pipeline is reasonably well supported by ablations.
3. The information-gain-based DPO is an interesting addition, especially because it tries to reward intermediate reasoning quality rather than only final outcomes.
4. The experiments are fairly comprehensive and show solid gains on two mobile-agent benchmarks.

**Weaknesses:**

1. It is not fully clear how much of the improvement comes from the architecture itself versus larger model size, extra training, or CoT-style supervision.
2. The novelty may be somewhat overstated: parts of the method look close to existing soft-MoE routing and prior information-gain-based training ideas.
3. The design is tightly tied to a fixed set of reasoning stages, which may make it harder to generalize beyond these specific GUI tasks.
4. The evaluation is limited to benchmark datasets, so it is still unclear how well the method works in more realistic online environments.
5. The reported gains over the strongest baselines are fairly small, so the “state-of-the-art” claim should be stated more carefully.

Reviewers acknowledged that the responses have addressed their concerns (even though one of the reviewers did not raise the score to above acceptance). Overall, there are no significant concerns remain unaddressed.